# Who can we trust? LLM-as-a-jury for Comparative Assessment

**Mengjie Qian** [1]   **Guangzhi Sun** [1]   **Mark J.F. Gales** [1]   **Kate Knill** [1]

## Abstract

Large language models (LLMs) are increasingly applied as automatic evaluators for natural language generation assessment often using pairwise comparative judgements. Existing approaches typically rely on single judges or aggregate multiple judges assuming equal reliability. In practice, LLM judges vary substantially in performance across tasks and evaluation aspects, and their judgment probabilities may be biased and inconsistent. Furthermore, human-labelled supervision for judge calibration may be unavailable. We first empirically demonstrate that inconsistencies in LLM comparison probabilities exist and show that it limits the effectiveness of direct probability-based ranking. To address this, we study the *LLM-as-a-jury* setting and propose BT-$\sigma$, a judge-aware extension of the Bradley-Terry model that introduces a discriminator parameter for each judge to jointly infer item rankings and judge reliability from pairwise comparisons alone. Experiments on benchmark NLG evaluation datasets show that BT-$\sigma$ consistently outperforms averaging-based aggregation methods, and that the learned discriminators strongly correlate with independent measures of the cycle consistency of LLM judgments. Further analysis reveals that BT-$\sigma$ can be interpreted as an unsupervised calibration mechanism that improves aggregation by modelling judge reliability.

## 1. Introduction

Large language models (LLMs) are increasingly used as automatic evaluators for reference-free assessment of natural language generation (NLG) tasks, including summarisation (Fabbri et al., 2021) and dialogue or story generation (Mehri & Eskenazi, 2020; Chhun et al., 2022; Gu et al., 2025).

[1]Department of Engineering, University of Cambridge, UK. Correspondence to: Mengjie Qian <mq227@cam.ac.uk>.

*Proceedings of the $43^{rd}$ International Conference on Machine Learning*, Seoul, South Korea. PMLR 306, 2026. Copyright 2026 by the author(s).

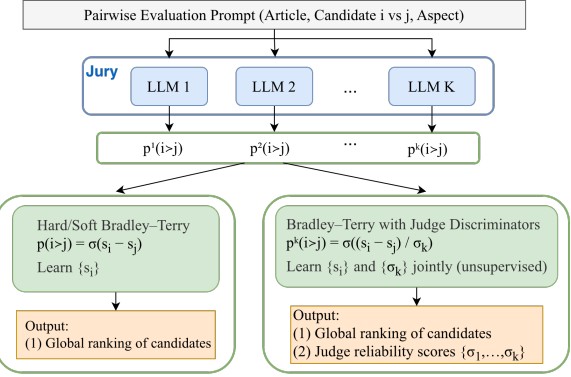

*Figure 1.* Hard/Soft Bradley-Terry and BT-$\sigma$ in the LLM-as-a-jury setting. Hard BT uses binary comparisons ($p(i \succ j) \in \{0, 1\}$), while soft BT uses probabilistic preferences. BT-$\sigma$ extends soft BT with judge-specific discriminators $\sigma_k$ to model judge reliability.

Among evaluation paradigms, pairwise comparative assessment has gained popularity due to its greater stability and cognitive grounding compared to absolute scoring, and is now widely adopted under the LLM-as-a-judge framework, where an LLM determines which of two candidates is better along a given aspect (Li et al., 2024). Despite its effectiveness, this approach assumes that judges are reliable, an assumption that often fails in practice: individual LLMs vary in evaluation quality and consistency, and their judgments are susceptible to systematic biases and prompt sensitivity. Prior work has identified issues such as self-preference bias and verbosity bias, as well as sensitivity to prompt phrasing and presentation choices, all of which can undermine evaluation reliability (Zheng et al., 2023; Stureborg et al., 2024; Wang et al., 2024; Verga et al., 2024). Treating heterogeneous judges as equally reliable can therefore lead to suboptimal aggregation outcomes.

To mitigate these issues, several approaches have been proposed to aggregate judgments from multiple LLMs. The most common strategy is simple averaging of probabilities or votes, which implicitly assumes uniform judge reliability (Verga et al., 2024; Badshah & Sajjad, 2025). Supervised calibration methods, such as temperature scaling (Guo et al., 2017), adjust probability outputs using labelled data but require human supervision and do not address differences in judge consistency. More recently, methods such as SkillAggregation (Sun et al., 2025) learns LLM-dependent weights

in a reference-free manner using neural models. While effective, such approaches rely on task-specific encoders and objectives, and do not explicitly model the probabilistic structure of pairwise comparisons.

In this work, we address these limitations by introducing a probabilistic framework, *BT-$\sigma$*, for *LLM-as-a-jury* that jointly models item rankings and judge reliability from pairwise comparisons alone. While Bradley-Terry models are well established (Bradley & Terry, 1952; Zermelo, 1929), their use for unsupervised reliability modelling of multiple LLM judges has not been systematically studied. Rather than assuming all judges are equally trustworthy or relying on calibrated probabilities, *BT-$\sigma$* jointly infers a global ranking of items and judge-specific reliability parameters directly from pairwise comparisons, without access to human-labelled supervision (Figure 1). This formulation addresses systematic inconsistencies and biases in LLM judgments and enables robust aggregation across heterogeneous evaluators. Main contributions are summarised as follows:

- **Diagnosis of inconsistency in LLM judgments.** We empirically show that pairwise probabilities produced by LLM judges often violate global ranking consistency, explaining why probability calibration alone is insufficient for reliable ranking recovery.

- **Judge-aware probabilistic aggregation** for comparative assessment. We introduce *BT-$\sigma$*, a judge-aware extension of the Bradley-Terry model that jointly learns item rankings and judge reliability from pairwise comparisons. To our knowledge, this is the first systematic study of unsupervised reliability-aware aggregation across multiple LLM judges in comparative NLG evaluation.

- **Unsupervised reliability modelling with strong empirical gains.** *BT-$\sigma$* enables unsupervised reliability-aware aggregation by calibrating judge behaviour without access to human labels, and consistently outperforms unweighted averaging and supervised temperature scaling.

## 2. Related Work

### 2.1. NLG Evaluation with LLMs

Evaluation for natural language generation (NLG) has historically relied on reference-based automatic metrics (Papineni et al., 2002; Lin, 2004; Banerjee & Lavie, 2005; Zhang et al., 2020; Zhao et al., 2019) and reference-free QA-style metrics such as QuestEval (Scialom et al., 2021). Recent advances in instruction tuning and alignment have endowed large language models (LLMs) with strong instruction-following abilities (Ouyang et al., 2022; Chung et al., 2024), enabling both proprietary systems (OpenAI Team, 2025; Gemini Team, 2025) and open-weight models (Grattafiori et al., 2024; Yang et al., 2025; DeepSeek-AI, 2024) to generate high-quality responses across diverse tasks. These

capabilities have motivated work that leverages LLMs as reference-free evaluators for NLG. Early approaches include GPTScore (Fu et al., 2024), which evaluates outputs via the conditional likelihood assigned by an LLM, while subsequent work has explored prompting-based LLM-as-a-judge paradigms (Li et al., 2024; Gu et al., 2025) that directly elicit scalar scores, rubric-based ratings, or structured critiques (Zheng et al., 2023; Chiang & yi Lee, 2023; Kocmi & Federmann, 2023; Liu et al., 2023; Wang et al., 2023).

To mitigate single-model biases and improve evaluation stability, multiple LLM judges, or LLM-as-a-jury, have been explored recently. Panel of LLM evaluators (PoLL) (Verga et al., 2024) shows that a diverse panel of smaller judges can outperform a single large judge while reducing intra-model bias. The Language Model Council (Zhao et al., 2025) has LLMs generate prompts, answer them, and then democratically evaluate one another to produce rankings on subjective tasks. More recent work, such as LLM Jury-on-Demand (Li et al., 2025b), CrossCheckGPT (Sun et al., 2024), and SkillAggregation (Sun et al., 2025), explores adaptive juries to dynamically select and weight judges per instance using reliability predictors. Classical EM-based (Dempster et al., 1977) approaches for annotator aggregation (e.g., Dawid-Skene-style formulations (Dawid & Skene, 1979)) also model annotator reliability, but typically assume repeated annotations and latent ground-truth labels, making them less suitable for reference-free LLM-as-a-jury settings with sparse pairwise comparisons. A closely related method from the crowdsourcing literature is Crowd-BT (Chen et al., 2013), which extends the Bradley-Terry model with a per-annotator reliability parameter modelled as a mixture between a skilled and a random annotator. However, Crowd-BT is restricted to binary decisions and uses a mixture-based mechanism for modelling reliability, making it fundamentally different from our proposed method.

### 2.2. LLM Comparative Assessments

More recent studies have refined the LLM-as-a-judge paradigm by emphasising pairwise comparisons and preference modelling, which tend to be more stable and better aligned with human judgments than absolute scoring (Wang et al., 2023; Liusie et al., 2024b). As a result, preference-style benchmarks and Arena-driven leaderboards became increasingly popular (Chiang et al., 2024; Li et al., 2025a; Bai et al., 2024). These approaches typically rely on a single LLM judge to produce pairwise preferences, implicitly assuming that the judge is reliable and consistent. Several works have explored reducing the computational cost of comparative assessment. Wang et al. (2023) study pairwise comparisons using both exhaustive and sorting-based strategies, while Park et al. (2024) apply comparative assessment to dialogue evaluation using randomly sampled comparisons and supervised adaptation of the judge model. More

broadly, probabilistic modelling of pairwise comparisons has been explored for ranking generated text. In particular, the Product of Experts (PoE) framework was proposed to obtain rankings of generated text efficiently by directly modelling the distribution of scores given a set of comparisons (Liusie et al., 2024c). Different to PoE, our work employs multiple LLM judges and estimates the reliability of each comparison from each judge to improve stability.

Recent work has also highlighted the pervasive cycle inconsistency problem in comparative assessments involving LLM judges. LLM-RankFusion (Zeng et al., 2024) shows that both order sensitivity and non-transitive comparisons undermine reliable ranking and addresses them via rank aggregation across multiple LLM rankers. Complementarily, Aligning with Logic (Liu et al., 2025) demonstrates that enforcing transitivity through preference refinement improves the coherence. TrustJudge (Wang et al., 2025) systematically identifies transitivity violations and preference cycles in LLM-as-a-judge settings and proposes probabilistic aggregation to mitigate these inconsistencies. Different from our setting, all the above methods employ a single judge for comparative assessments.

## 3. Comparative Assessment

### 3.1. Bradley-Terry Models

In pairwise comparative assessment, an evaluator is asked to decide which of two candidates is better along a given aspect. A common way to aggregate such judgments is to count win-loss ratios or to average pairwise preference probabilities across comparisons. While simple, these approaches treat each comparison independently and do not impose any global consistency constraints, often resulting in unstable or incoherent rankings.

A principled probabilistic model for pairwise comparisons is the Bradley-Terry (BT) model. Given two items $i$ and $j$, BT defines the probability that $i$ is preferred over $j$ as

$$P(i \succ j) = p_{ij} = \sigma(s_i - s_j), \quad (1)$$

where $s_i$ and $s_j$ are latent skill parameters and $\sigma(x) = \frac{1}{1+e^{-x}}$ is the logistic function. In its standard form, the BT model assumes binary outcomes and estimates item skills by maximising the likelihood of observed win-loss comparisons:

$$\mathcal{L}_{\text{hard}}(\mathbf{s}) \propto \prod_{(i,j)\in\mathcal{C}} \sigma(s_i - s_j)^{y_{ij}}\big(1 - \sigma(s_i - s_j)\big)^{1-y_{ij}}, \quad (2)$$

where $y_{ij} \in \{0, 1\}$ is the binary comparison, $\mathcal{C}_{1:M}$ denotes the set of pairwise comparisons and $\mathbf{s}$ is the set of latent skills. We refer to this formulation as **hard BT**.

In LLM-based evaluation, however, judges typically produce soft preference probabilities rather than binary deci-

sions. To leverage this richer information, recent work has adopted a soft Bradley-Terry formulation, also known as *PoE-BT* (Liusie et al., 2024c), in which observed probabilities are treated as noisy measurements of an underlying BT model. Specifically, given a set of pairwise comparisons $\mathcal{C}$ with observed probabilities $p_{ij}$, the likelihood is defined as

$$\mathcal{L}_{\text{soft}}(\mathbf{s}) \propto \prod_{(i,j)\in\mathcal{C}} \sigma(s_i - s_j)^{p_{ij}}\big(1 - \sigma(s_i - s_j)\big)^{1-p_{ij}}. \quad (3)$$

This formulation reduces to *hard BT* when $p_{ij} \in \{0, 1\}$ and is therefore a strict generalisation. We refer to this model as **soft BT**. The key distinction between *hard BT* and *soft BT* is therefore whether the model is fitted to binary outcomes ($y_{ij} \in \{0, 1\}$) or soft probabilities ($p_{ij} \in [0, 1]$).

**Debiasing pairwise probabilities.** Directly applying BT models to raw LLM probabilities can be problematic due to systematic positional bias. In practice, LLM judgments may violate *commutativity* due to *positional bias*, producing asymmetric probabilities such that $p_{ij} + p_{ji} \neq 1$, depending on candidate order or prompt formatting. This behaviour is a known form of logical inconsistency in LLM-based comparative assessment (Liusie et al., 2024a; Liu et al., 2025). To mitigate this issue, we apply a simple symmetrisation step before fitting any BT model. For each pair $(i, j)$, we define a debiased probability

$$p'_{ij} = \tfrac{1}{2}\big(p_{ij} + (1 - p_{ji})\big), \quad (4)$$

which enforces commutativity by construction. For *hard BT*, binary outcomes are then obtained as $y_{ij} = \mathbb{I}[p'_{ij} \geq 0.5]$. This debiasing step is applied consistently across all BT-based methods in this paper and ensures that observed preferences satisfy a minimal structural assumption required by the BT model.

### 3.2. Probability Consistency and Calibration

**Probability consistency and self-calibration.** Under the *soft BT* formulation (Eq. (3)), item skills are estimated by maximising the log-likelihood given observed pairwise probabilities. The stationarity condition of the log-likelihood with respect to item skills yields

$$\nabla \log \mathcal{L}_{\text{soft}}(\mathbf{s}) = \sum_{(i,j)\in\mathcal{C}_{1:M}} \big(p_{ij} - \sigma(s_i - s_j)\big) = 0. \quad (5)$$

This equation shows that *soft BT* estimates skills by matching the model-predicted probabilities $\sigma(s_i - s_j)$ to the observed pairwise probabilities $p_{ij}$ in expectation. When the probabilities are internally consistent with BT, i.e. $p_{ij} = \sigma(s_i - s_j)$ for some latent skills $\mathbf{s}$, this condition is satisfied up to a global shift of the skill space. In this ideal case, applying temperature scaling to the probabilities is equivalent to a global rescaling of the skill space and therefore does

not change the induced ranking (Fathullah & Gales, 2025):

$$p_{ij} = \sigma(s_i - s_j) \iff$$

$$\tilde{p}_{ij} = \frac{p_{ij}^{1/T}}{p_{ij}^{1/T} + (1 - p_{ij})^{1/T}} = \sigma(s_i - s_j) \iff$$

$$\tilde{p}_{ij} = \sigma\big((s_i - s_j)/T\big) \qquad (6)$$

where $\tilde{p}_{ij}$ is the temperature-annealed probability. As a result, in this ideal case *soft BT* will implicitly perform *self-calibration* with temperature scaling. *Hard BT* can be viewed as an extreme case of this process, corresponding to a very small temperature that retains only the sign of each comparison. Under probability consistency, *hard BT* and *soft BT* therefore recover identical rankings.

In practice, however, LLM-generated probabilities often violate logic consistency, such as transitivity, commutativity, and negation invariance (Liu et al., 2025). These inconsistencies make it difficult for the BT model to identify a single skill vector that can fully explain the observed probabilities. Under such conditions, *soft BT*, which relies on soft probabilities, must reconcile conflicting probability signals and can potentially amplify noise and degrade ranking quality. *Hard BT*, by discarding probability magnitudes and retaining only comparison directions, can act as a more robust estimator when probabilities are highly inconsistent. This provides a principled explanation for why, in empirical experiments, *hard BT* can outperform *soft BT* on some evaluation aspects when using individual LLM judges.

**Cycle inconsistency rate.** In this work, we also systematically evaluate one common form of probability inconsistency, i.e. the violation of transitivity. For a triplet of items $(i, j, k)$, a cycle inconsistency occurs if the preferences form a directed loop, i.e. $i \succ j$, $j \succ k$, and $k \succ i$. The prevalence of such cycles indicates the degree to which a judge's probabilities deviate from any latent skill-based model. Violations of transitivity are formalised as *pairwise transitivity inconsistency* in prior work (Wang et al., 2025), which characterises inconsistency via directed cycles over subsets of size $k \geq 3$. In this paper, we focus on the $k = 3$ case and quantify transitivity inconsistency using the rate of directed 3-cycles. To quantify this behaviour, we define a *cycle inconsistency rate* (CycleRate, Eq. (9)). Given pairwise preference probabilities $p_{ij}$, the adjacency matrix $A \in \{0, 1\}^{n \times n}$ is defined such that

$$A_{ij} = \begin{cases} 1, & \text{if } p_{ij} > 0.5, \\ 0, & \text{otherwise.} \end{cases} \qquad (7)$$

Here, we focus on cycles of length three, noting that longer cycles may also exist. A cycle is present for a triplet $(i, j, k)$ if either orientation of a directed 3-cycle occurs. The total number of such cycles is computed as

$$N_{\text{cycles}} = \sum_{i < j < k} \Big[ \mathbb{1}\big(A_{ij} A_{jk} A_{ki} = 1\big) + \mathbb{1}\big(A_{ik} A_{kj} A_{ji} = 1\big) \Big]. \qquad (8)$$

Let $N_{\text{triples}} = \binom{n}{3}$ denote the total number of triplets. The cycle inconsistency rate is then defined as

$$\text{CycleRate} = \frac{N_{\text{cycles}}}{N_{\text{triples}}}. \qquad (9)$$

Higher-order cycles ($k \geq 4$) correspond to more complex compositions of pairwise inconsistencies. They can be decomposed into 3-cycles, so making 3-cycle analysis a representative measure of overall inconsistency. To validate this empirically, 4-cycle rates are computed across all judges on SummEval and found to be strongly correlated with 3-cycle rates, confirming that 3-cycle analysis captures the dominant inconsistency signal.

### 3.3. Reliability-Aware Aggregation

To improve robustness, multiple LLMs are often used as judges and their pairwise judgments are aggregated. We refer to this setting as *LLM-as-a-jury*. A common aggregation strategy is to average probabilities across judges before applying a ranking model. This approach assumes that all judges are equally reliable and implicitly applies a single global calibration, despite the fact that different LLMs may exhibit different levels of noise and inconsistency.

Moreover, applying *soft BT* directly to probabilities from all LLM judges is equivalent to first averaging all judges' probabilities then applying *soft BT*. Formally, let $p_{ij}^{(k)}$ denote the preference probability from judge $k$ for item $i$ over $j$. The *soft BT* likelihood over all judges can be written as

$$\mathcal{L}_{\text{soft}}^{\text{jury}}(\mathbf{s}) \propto \prod_{k=1}^{K} \prod_{(i,j) \in \mathcal{C}} \sigma(s_i - s_j)^{p_{ij}^{(k)}} \big(1 - \sigma(s_i - s_j)\big)^{1 - p_{ij}^{(k)}}. \qquad (10)$$

Taking derivatives with respect to the item skills yields the stationarity condition

$$\nabla \log \mathcal{L}_{\text{soft}}^{\text{jury}}(\mathbf{s}) = \sum_{k=1}^{K} \sum_{(i,j) \in \mathcal{C}_{1:M}} \big(p_{ij}^{(k)} - \sigma(s_i - s_j)\big) \qquad (11)$$

$$\propto \sum_{(i,j) \in \mathcal{C}_{1:M}} \left( \frac{1}{K} \sum_{k=1}^{K} p_{ij}^{(k)} - \sigma(s_i - s_j) \right) = 0,$$

where $\frac{1}{K} \sum_{k=1}^{K} p_{ij}^{(k)}$ denotes the averaged probabilities across all LLM judges for the pair $(i, j)$. Eq. (11) shows that, in the jury setting, *soft BT* matches model predictions to the average probability across judges. As a result, *soft BT* over multiple judges can only learn a single global ranking and a single implicit calibration shared by all judges.

This formulation cannot account for heterogeneity in judge reliability or inconsistency, and may therefore limit aggregation performance when some judges produce substantially noisier or less consistent probabilities than others.

One natural solution is to calibrate each judge independently, for example, using temperature scaling before aggregation. However, estimating an optimal temperature for each judge requires labelled data and incurs additional computational cost, making this approach impractical in many reference-free evaluation settings.

We propose an alternative solution that integrates reliability modelling directly into the aggregation process. Specifically, we extend the *soft BT* model (Eq. (3)) by introducing a judge-specific discriminator $\sigma_k$ for each judge $k$:

$$\mathcal{L}(\mathbf{s}, \{\sigma_k\}) \propto \prod_{k=1}^{K} \prod_{(i,j)\in\mathcal{C}} \sigma\left(\frac{s_i - s_j}{\sigma_k}\right)^{p_{ij}^{(k)}} \left(1 - \sigma\left(\frac{s_i - s_j}{\sigma_k}\right)\right)^{1-p_{ij}^{(k)}},$$
(12)

where $\sigma_k$ controls how sensitive judge $k$'s probabilities are to underlying skill differences. Smaller values correspond to more discriminative and internally consistent judges, while larger values indicate noisier or less reliable probability estimates. The parameters $\{s_i\}$ and $\{\sigma_k\}$ are learned jointly by maximising the likelihood over all judges and comparisons, without human labels. We denote this model as **BT-$\sigma$**.

The discrimination parameter $\sigma_k$ is closely related to temperature scaling. For a fixed judge, it rescales the effective margin between items in the same way a temperature rescales logits. The key distinction is that temperature scaling operates on raw model outputs and requires labelled data, whereas $\sigma_k$ is learned from pairwise comparisons alone without access to human annotations. In this sense, $BT$-$\sigma$ can be viewed as an unsupervised, comparative analogue of calibration. This equivalence holds when the judge's probabilities are perfectly consistent with the BT model, i.e. $p_{ij} = \sigma(s_i - s_j)$; under inconsistency the two approaches diverge, as temperature scaling adjusts probability magnitudes against human labels whereas $\sigma_k$ absorbs structural inconsistency directly during ranking inference. Importantly, this $\sigma_k$ is meaningful only in the multi-judge, soft-comparison setting. In single-judge or hard BT models, a global scaling factor $\sigma_k$ can be absorbed into the item skills and becomes uninformative. When modelling soft probabilities from multiple judges, however, relative differences in $\sigma_k$ across judges provide a meaningful signal of judge reliability.

Given that most NLG datasets have different evaluation aspects, we also consider an aspect-dependent variant, **BT-$\sigma$-asp**, which estimates a separate discriminator for each judge-aspect pair. This extension allows reliability to vary across evaluation dimensions. Empirically, we find that a single discriminator per judge is often sufficient, suggesting that judge reliability is largely stable across aspects.

## 4. Experimental Setup

**Datasets.** Experiments are conducted on three widely used benchmark NLG datasets. **SummEval** (Fabbri et al., 2021) contains 100 articles from the CNN/DailyMail dataset, each associated with 16 machine-generated summaries. Each summary is annotated with human scores on four aspects: coherence (COH), consistency (CON), fluency (FLU) and relevance (REL). Pairwise preferences are derived from these human scores. In this work, SummEval serves as the primary benchmark for evaluating aggregation quality and judge reliability. **Topical-Chat** (Mehri & Eskenazi, 2020) is a dialogue response generation dataset consisting of 60 dialogue contexts, each associated with 6 candidate responses. Human annotations are provided for four dialogue-specific aspects: coherency (COH), continuity (CNT), engagingness (ENG), and naturalness (NAT). This dataset is used to evaluate whether the proposed aggregation method generalises beyond summarisation to dialogue response evaluation. **NovelEval** (Sun et al., 2023) is a question-answering benchmark for evaluating passage relevance, consisting of 21 novel questions published after the release of GPT-4, with relevance (REL) as the single evaluation aspect. Following prior work (Liu et al., 2025), NovelEval is used as a further benchmark to evaluate the generalisation of our approach. We exclude samples with near-zero average win probability (Avg-Prob) Spearman's rank correlations (SRC), where all judges produce near-random preferences, and evaluate on the retained subset.

**LLM judges and prompts** are detailed in Appendix D.

**Baseline and proposed approaches.** We evaluate the following methods. *Avg-Prob* corresponds to directly using the model's pairwise preference probabilities, by averaging each item's win probability across all comparisons, i.e. $w_i = \frac{1}{N-1} \sum_{j \neq i} p_{ij}$. This approach does not enforce any global ranking structure, and serves as the primary baseline. *Hard BT* and *soft BT* fit a Bradley-Terry model to binary and soft pairwise preferences respectively, as described in Section 3.1. *Temp-BT* applies supervised temperature scaling to each judge's probabilities before fitting a soft BT model, where a separate temperature is estimated for each judge on each evaluation aspect by minimising expected calibration error (ECE) against human judgments. Temperatures are fit directly on the evaluation data due to the lack of a held-out development set, *Temp-BT* is therefore included only as a reference point. The proposed **BT-$\sigma$** and **BT-$\sigma$-asp** extend *soft BT* with judge-specific and judge-aspect-specific discriminators respectively, as described in Section 3.3. *Hard BT-$\sigma$* applies judge-specific discriminators to binary comparisons rather than soft probabilities. *Crowd-BT* (Chen et al., 2013), discussed in Section 2.1, extends the Bradley-Terry model with a mixture-based per-judge reliability parameter and is included as an additional

reliability-aware baseline.

**Implementation details.** BT-$\sigma$ is optimised by maximising the joint log-likelihood over item skills $\{s_i\}$ and judge discriminators $\{\sigma_k\}$ using the L-BFGS-B method (Zhu et al., 1997; Morales & Nocedal, 2011) via `scipy`, which adapts step sizes internally with a quasi-Newton Hessian approximation and typically converges within 100 iterations. Parameters $\{s_i\}$ and $\{\sigma_k\}$ are initialised with random values drawn from a uniform distribution over $[0, 1)$.

**Evaluation.** We use Spearman's rank correlations (SRC) (Wissler, 1905) between human scores and LLM judge ranking as the assessment metric. SRC captures relative ordering consistency and is widely used in reference-free evaluation settings. Higher SRC indicates closer alignment with human preferences. For all three benchmark datasets, LLM judges are used to produce pairwise preference probabilities for all possible comparisons within each article or context, resulting in $N(N-1)$ comparisons for each article or context, where $N$ is the number of candidate summaries or responses. We calculate the SRC on each article or context then average to get the overall score.

## 5. Results

### 5.1. Bradley-Terry Performance and Probability Inconsistency

Before considering aggregation, we first analyse the behaviour of BT models when applied to pairwise probabilities produced by individual LLM judges. Table 1 reports SRC between rankings predicted by different methods and human judgments on different aspects of SummEval, i.e. coherence (COH), consistency (CON), fluency (FLU), and relevance (REL). The ALL column reports the overall performance, measured by averaging over the four aspects. Here, Avg-Prob is used as baseline.

Across most models and aspects, both soft BT and hard BT outperform direct averaging of probabilities (Avg-Prob), indicating that enforcing a global ranking structure improves robustness over treating comparisons independently. Llama-3.2-3B is the only clear exception: the overall SRC is much lower than other models, and the SRC on CON and FLU are particularly low, suggesting that this judge produces highly unreliable or near-random preference signals. Excluding this outlier, we observe that hard BT often matches or exceeds the performance of soft BT, and in several cases yields substantially higher SRC (e.g., Phi-3.5-mini-instruct on COH, Qwen-2.5-7B-Instruct on COH, DeepSeek-LLM-7B-Chat on REL). As discussed in Section 3.2, under ideal conditions where pairwise probabilities are internally consistent, soft BT and hard BT should recover identical rankings, since hard BT corresponds to an extreme temperature scaling of soft BT. The empirical gap between hard and soft BT

*Table 1.* Spearman correlations (SRC) of each individual LLM judge on SummEval. The best results for each aspect on each model are underlined.

| Model | Method | COH | CON | FLU | REL | ALL |
|---|---|---|---|---|---|---|
| Llama-3.1-8B | Avg-Prob | 43.87 | 31.89 | 26.32 | 42.39 | 36.12 |
| | hard BT | 46.88 | 45.47 | 37.09 | 50.69 | 45.03 |
| | soft BT | 48.40 | 45.12 | 36.80 | 51.02 | 45.33 |
| Llama-3.2-3B | Avg-Prob | 27.72 | 5.93 | 7.12 | 30.61 | 17.84 |
| | hard BT | 13.31 | 0.28 | 17.24 | 33.15 | 16.00 |
| | soft BT | 13.36 | 0.53 | 14.50 | 34.40 | 15.70 |
| Mistral-7B-Instruct-v0.3 | Avg-Prob | 33.65 | 32.08 | 20.99 | 36.04 | 30.69 |
| | hard BT | 32.49 | 39.65 | 25.02 | 39.45 | 34.15 |
| | soft BT | 34.42 | 36.44 | 25.32 | 40.07 | 34.06 |
| Phi-3.5-mini-instruct | Avg-Prob | 40.50 | 43.08 | 38.10 | 44.88 | 41.64 |
| | hard BT | 47.01 | 44.89 | 39.57 | 50.46 | 45.48 |
| | soft BT | 42.04 | 44.46 | 39.58 | 48.17 | 43.56 |
| Qwen2.5-3B-Instruct | Avg-Prob | 42.24 | 40.47 | 34.83 | 44.94 | 40.62 |
| | hard BT | 46.89 | 46.93 | 39.97 | 47.10 | 45.22 |
| | soft BT | 46.51 | 45.63 | 39.00 | 46.48 | 44.40 |
| Qwen2.5-7B-Instruct | Avg-Prob | 42.74 | 36.94 | 31.39 | 47.70 | 39.69 |
| | hard BT | 51.42 | 43.35 | 33.61 | 50.04 | 44.61 |
| | soft BT | 44.79 | 40.95 | 32.39 | 51.47 | 42.40 |
| DeepSeek-LLM-7b-Chat | Avg-Prob | 27.70 | 25.49 | 20.71 | 30.40 | 26.08 |
| | hard BT | 32.91 | 34.96 | 34.27 | 42.30 | 36.11 |
| | soft BT | 30.65 | 31.96 | 32.47 | 37.79 | 33.22 |
| Gemma-2-9B-IT | Avg-Prob | 60.67 | 39.74 | 34.45 | 52.46 | 46.83 |
| | hard BT | 61.57 | 44.71 | 38.96 | 54.68 | 49.98 |
| | soft BT | 61.20 | 44.92 | 40.20 | 55.30 | 50.40 |
| All models | Avg-Prob | 52.55 | 41.75 | 36.21 | 50.09 | 45.15 |
| | hard BT | 51.26 | 45.72 | 40.07 | 52.32 | 47.34 |
| | soft BT | 53.94 | 47.86 | 42.69 | 53.11 | 49.40 |

therefore indicates violations of probability consistency in LLM judgments.

To quantify internal inconsistency, Figure 2 plots the cycle inconsistency rate (Eq. (9)) for each model on the COH aspect. Models with higher cycle inconsistency rates, indicating frequent violations of transitivity, are generally those for which hard BT yields larger gains over soft BT. Mistral-7B-Instruct-v0.3 forms a notable exception: although its cycle inconsistency is high, soft BT performs better on COH. This suggests that cycle inconsistency alone does not fully determine the relative effectiveness of hard and soft BT, and that additional factors may influence how probability noise interacts with the ranking model. We leave a more detailed analysis of this behaviour to future work. In contrast, models whose inconsistencies are more systematic can benefit more from hard BT, which performs as a robust estimator by retaining only comparison directions. Cycle inconsistency results for other aspects are provided in Appendix A.1.

When judgments from multiple LLMs are aggregated, individual inconsistencies tend to average out, reducing the

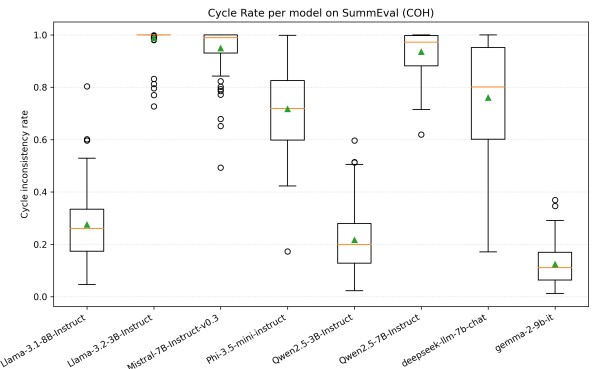

*Figure 2.* Cycle inconsistency rate on different LLMs, evaluated on SummEval (COH).

impact of extreme probability noise. As a result, soft BT becomes more effective in the aggregated setting and typically outperforms hard BT, as shown in the bottom rows of Table 1. This behaviour highlights the complementary roles of hard and soft BT: hard BT is robust under severe probability inconsistency, whereas soft BT performs better when inconsistency is moderate or low.

Results on Topical-Chat exhibit similar trends for individual models, though inconsistency levels are generally higher than on SummEval. In the aggregated setting, soft BT outperforms hard BT on most aspects, though the margin is smaller than on SummEval. It still underperforms on ENG, where probability noise remains substantial (with a CycleRate of 0.44). Detailed results and inconsistency analyses on Topical-Chat are reported in Appendix B.1 and B.2.

Further experiments on NovelEval support the generalisation of these observations, with results and discussion reported in Appendix B.3.

### 5.2. Reliability-Aware Aggregation with BT-$\sigma$

We now evaluate aggregation performance when combining pairwise judgments from multiple LLM judges. Figures 3a and 3b summarise the overall SRC performance across methods on all aspects for SummEval and Topical-Chat, respectively. Table 2 reports SRC on each aspects for the two datasets. Across both datasets, Avg-Prob provides a reasonable baseline but consistently underperforms methods that impose a global ranking structure, confirming that enforcing transitivity and global consistency is beneficial when combining noisy judgments. However, differences emerge between hard BT, soft BT, and reliability-aware variants.

On SummEval, soft BT consistently outperforms hard BT across all aspects, indicating that aggregation reduces extreme probability noise and allows soft probability information to be exploited effectively. On Topical-Chat, soft BT achieves stronger overall performance than hard BT, but

underperforms on ENG, suggesting the probability inconsistency on ENG remains substantial even after aggregation, making magnitude-based methods less reliable in this setting. Further analysis shows that the average CycleRate on ENG is 0.44 across all models, aligning with this observation. These contrasting behaviours reinforce the analysis in Section 3.2 that soft BT benefits under moderate inconsistency, whereas hard BT is effective under severe noise.

The proposed BT-$\sigma$ consistently exceeds soft BT on both datasets. By introducing a judge-specific discriminator, BT-$\sigma$ downweights unreliable probability signals and effectively performs reliability-aware aggregation. This allows it to retain the advantages of soft BT when probabilities are informative, while mitigating the impact of highly inconsistent judges. In the case of weak individual judges and noisy probabilities (such as ENG on Topical-Chat), where soft BT lags behind hard BT, BT-$\sigma$ yields more effective aggregation and surpasses hard BT, as shown on Topical-Chat (Table 2).

We also evaluate an aspect-dependent variant of the BT model (BT-$\sigma$-asp), which estimates a separate discrimination parameter for each judge-aspect pair. This variant is intended to test whether an LLM's reliability varies across evaluation dimensions. BT-$\sigma$-asp yields only marginal improvements over BT-$\sigma$ on overall performance for both SummEval and Topical-Chat (Figures 3a and 3b). A closer look at per-aspect results in Table 2 shows modest but consistent gains on SummEval, while improvements on Topical-Chat are mixed and aspect-dependent. These results suggest that LLM judge reliability can vary slightly across evaluation aspects, though the differences are relatively small in practice. Given the additional parameters and training cost required by aspect-specific discriminators, the simpler BT-$\sigma$ model appears to be sufficient for most practical applications, offering a balance between performance and efficiency.

Another variant of our method, hard BT-$\sigma$, which applies judge-specific discriminators $\sigma_k$ to binary (hard) comparisons rather than soft probabilities, is evaluated. This variant is motivated by the observation that the learned $\sigma_k$ values correlate strongly with cycle inconsistency (see detailed analysis in Section 5.3), while hard BT alone, which ignores probability magnitudes and relies solely on comparison directions, cannot mitigate such structural violations. As shown in Figure 3b, hard BT-$\sigma$ is particularly effective on Topical-Chat, where inconsistency levels are high, outperforming both soft BT and soft BT-$\sigma$ on several aspects (Table 2); these are also the aspects exhibiting the largest cycle rates. In contrast, on SummEval, where inconsistencies are more moderate, soft BT-$\sigma$ remains superior. These results suggest a clear pattern: when probability signals are highly inconsistent, adding reliability modelling to hard comparisons (hard BT-$\sigma$) yields the most robust aggregation, whereas under moderate inconsistency, reliability-aware soft

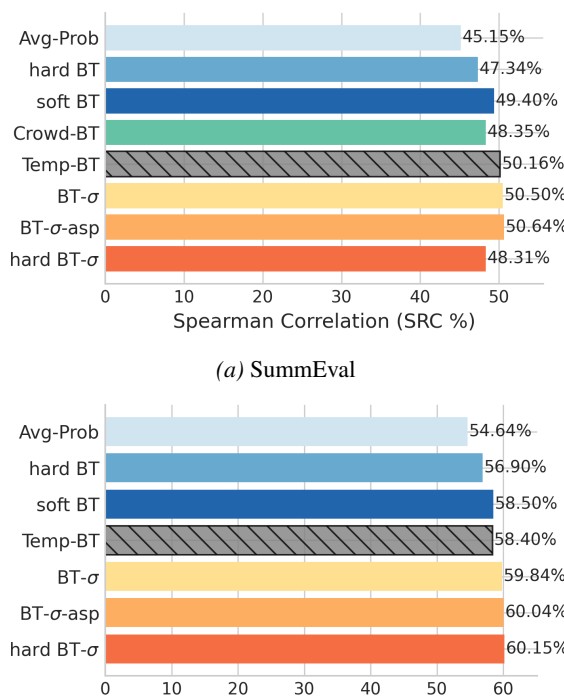

*(a)* SummEval

*(b)* Topical-Chat.

*Figure 3.* SRC barchart for different aggregation methods on SummEval and Topical-Chat (ALL aspects).

*Table 2.* Aggregation performance measured by Spearman correlations (SRC) between aggregated rankings and human judgments, evaluated on SummEval and Topical-Chat.

| Method | SummEval | | | | Topical-Chat | | | |
|---|---|---|---|---|---|---|---|---|
| | COH | CON | FLU | REL | COH | CNT | ENG | NAT |
| Avg-Prob | 52.55 | 41.75 | 36.21 | 50.09 | 56.01 | 49.39 | 61.62 | 51.53 |
| hard BT | 51.26 | 45.72 | 40.07 | 52.32 | 59.31 | 50.25 | 62.57 | 56.90 |
| soft BT | 53.94 | **47.86** | 42.69 | 53.11 | 60.05 | 53.87 | 61.87 | 58.20 |
| Temp-BT | 56.21 | 47.40 | 41.88 | **55.14** | 56.88 | 52.21 | 63.86 | 60.65 |
| BT-$\sigma$ | **57.38** | 47.47 | 42.99 | 54.15 | 59.02 | **56.30** | 63.49 | 60.56 |
| BT-$\sigma$-asp | 57.36 | 47.56 | **43.08** | 54.56 | 58.94 | 54.92 | 65.58 | **60.71** |
| hard BT-$\sigma$ | 53.02 | 47.08 | 40.44 | 52.69 | **60.89** | 53.45 | **67.36** | 58.90 |

Crowd-BT is evaluated on SummEval. As shown in Figure 3a, Crowd-BT achieves an overall SRC of 48.35, outperforming hard BT (47.34) but under-performing soft BT (49.40) and BT-$\sigma$ (50.50), suggesting that the mixture-based reliability mechanism of Crowd-BT provides limited gains over standard and our proposed BT aggregation in the LLM soft probability setting. Crowd-BT results are therefore not reported on Topical-Chat.

Results on an additional benchmark, NovelEval, show consistent improvements of BT-$\sigma$ over other approaches, further supporting the generalisation of the proposed approach (Appendix B.3).

### 5.3. Judge Reliability Analysis

This section examines whether the judge-specific discriminators learned by BT-$\sigma$ capture meaningful differences in LLM judge reliability. Intuitively, a smaller $\sigma_k$ (equivalently, a larger $1/\sigma_k$) corresponds to a judge with sharper and more consistent pairwise judgments. Figure 4 plots $1/\sigma_k$ against the evaluation performance of each LLM judge on SummEval, measured by SRC between Avg-Prob rankings and human judgments. A clear positive relationship is observed, with higher-performing judges exhibiting smaller discriminator values (i.e. larger $1/\sigma_k$). The overall Pearson correlation (PCC) is 72.2%, suggesting that the learned discriminator $\sigma_k$ provides a strong unsupervised signal of judge reliability. A similar trend is observed on Topical-Chat and NovelEval, and the corresponding scatter plots are provided in Appendix B.3 and Appendix C.2.

Table 3 reports both Pearson (PCC) and Spearman (SRC) correlations between $1/\sigma_k$ and judge performance across evaluation aspects on SummEval and Topical-Chat. Positive correlations are consistently observed across all aspects and both datasets. While PCC reflects linear alignment between discriminator magnitude and absolute performance, the consistently strong SRC values indicate that the learned discriminators reliably preserve the relative ordering of judges by quality. This alignment explains why BT-$\sigma$ improves

aggregation (BT-$\sigma$) remains the most effective approach.

Statistical significance tests have been conducted comparing the proposed methods against Avg-Prob. On SummEval, hard BT-$\sigma$ shows statistically significant improvements on all four aspects ($p < 0.001$ on COH and CON, and $p < 0.05$ on FLU and REL). BT-$\sigma$-asp and hard BT-$\sigma$ variants also show consistent significance. Results are more mixed on Topical-Chat: significance is reached on NAT ($p < 0.05$), while other aspects show consistent but not always significant gains. This aligns with our observation that Topical-Chat exhibits higher overall inconsistency levels, making aggregation inherently more difficult.

We also report results for Temp-BT, which uses human annotations to estimate the optimal temperature (see Section 4 for details). As shown in Figures 3a and 3b, Temp-BT improves over uncalibrated soft BT on SummEval, but does not offer consistent gains on Topical-Chat. In contrast, BT-$\sigma$ consistently exceeds Temp-BT across both datasets without access to human labels. This highlights a key distinction between supervised probability calibration and reliability-aware aggregation: while temperature scaling adjusts probability magnitudes to better match annotations, BT-$\sigma$ improves aggregation by explicitly modelling judge reliability in an unsupervised manner.

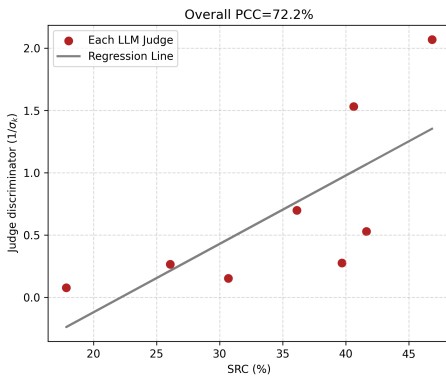

*Figure 4.* Scatter plot of $1/\sigma_k$ and LLM performance with Avg-Prob, measured by SRC, on SummEval (ALL).

*Table 3.* Correlation between learned discriminator $1/\sigma_k$ and LLM rankings (PCC/SRC, %) on SummEval and Topical-Chat.

| Dataset | Metric | COH | CON | FLU | REL | ALL |
|---------|--------|-----|-----|-----|-----|-----|
| SummEval | PCC | 84.11 | 54.05 | 60.75 | 71.92 | 72.21 |
| | SRC | 80.95 | 66.67 | 76.19 | 76.19 | 85.71 |
| | | COH | CNT | ENG | NAT | ALL |
| Topical-Chat | PCC | 58.68 | 73.01 | 59.79 | 70.09 | 67.41 |
| | SRC | 40.48 | 73.81 | 61.90 | 76.19 | 59.52 |

*Table 4.* Correlation between learned discriminator $1/\sigma_k$ and $1 - \text{CycleRate}$ (PCC/SRC, %) on SummEval and Topical-Chat.

| Dataset | Metric | COH | CON | FLU | REL | ALL |
|---------|--------|-----|-----|-----|-----|-----|
| SummEval | PCC | 90.17 | 90.33 | 85.72 | 88.25 | 90.29 |
| | SRC | 97.62 | 97.62 | 88.10 | 97.62 | 95.24 |
| | | COH | CNT | ENG | NAT | ALL |
| Topical-Chat | PCC | 90.17 | 89.99 | 90.41 | 81.61 | 93.09 |
| | SRC | 85.71 | 92.86 | 92.86 | 83.33 | 90.48 |

aggregation performance, as judges with higher internal consistency exert greater influence on the inferred ranking while noisier judges are naturally downweighted.

Table 4 reports the correlation between the learned discriminator $1/\sigma_k$ and cycle inconsistency (measured as $1 - \text{CycleRate}$), which reflects the internal consistency of LLM judges. Corresponding scatter plots for the two datasets on overall aspects are in Appendix A.2 and B.3. Across both SummEval and Topical-Chat, we observe strong positive correlations under both PCC and SRC metrics, consistently across individual aspects and overall scores. This indicates that judges assigned larger values of $1/\sigma_k$ tend to exhibit fewer transitivity violations, whereas noisier judges with higher cycle inconsistency are assigned lower effective weight (i.e., smaller $1/\sigma_k$). These results support the interpretation of $1/\sigma_k$ as an unsupervised measure of judge reliability and suggest that BT-$\sigma$ effectively captures and exploits differences in internal consistency among LLM judges during aggregation.

## 6. Conclusions

This paper studies the problem of aggregating pairwise judgments from multiple LLM judges of unknown reliability in reference-free comparative assessment. We show empirically that LLM-generated comparison probabilities are often internally inconsistent, which limits the effectiveness of probability-based aggregation and explains the empirical gap between soft and hard Bradley-Terry models. Motivated by this observation, we propose *BT-$\sigma$*, a judge-aware extension of the BT model that introduces a discriminator parameter to model differences in judge reliability and enables unsupervised aggregation directly from pairwise comparisons. Across multiple NLG evaluation benchmarks, *BT-$\sigma$* consistently improves aggregation performance over averaging-based and globally calibrated baselines. We further show that a hard-comparison variant, *hard BT-$\sigma$*, provides additional robustness when probability signals are highly inconsistent. The learned discriminator correlates strongly with independent measures of LLM evaluation performance, providing an interpretable signal of judge reliability. By explicitly modelling heterogeneous reliability rather than relying on probability calibration, *BT-$\sigma$* offers a principled and practical solution for LLM-as-a-jury aggregation under realistic conditions where probabilities are noisy, inconsistent, and unlabelled.

## 7. Limitations

**Reliance on token logits.** BT-$\sigma$ derives preference probabilities from token-level logits, which restricts the jury to open-weight models. Closed-source models do not expose logits, limiting the applicability of the method to settings where such access is available.

**Potential for systematic bias.** BT-$\sigma$ estimates judge reliability based on internal consistency rather than agreement with the majority, allowing highly consistent minority judges to be upweighted even if they disagree with other judges. However, if a majority of judges are both internally consistent and systematically biased (e.g. sharing a common preference for longer responses), this bias may persist in the aggregated ranking. This is a fundamental limitation of unsupervised aggregation. Without ground truth, it is not possible to distinguish a consistent biased majority from a correct minority.

## Acknowledgements

This paper reports on research supported by Cambridge University Press & Assessment, a department of The Chancellor, Masters, and Scholars of the University of Cambridge.

## Impact Statement

This work advances methods for reference-free evaluation of NLG by improving the robustness and interpretability of aggregating judgments from multiple LLMs. By enabling unsupervised modelling of judge reliability, the proposed approach can reduce the influence of unreliable or inconsistent evaluators and support more trustworthy automatic evaluation pipelines. While the method is intended for evaluation rather than deployment in decision-critical systems, it may indirectly affect how models are compared and selected. Aggregation methods can improve robustness, but improved agreement does not imply that shared or systematic biases across LLM judges have been removed.

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

# A. Plots on SummEval

## A.1. Cycle Inconsistency Rate on SummEval

This subsection reports the cycle inconsistency rates of individual LLM judges on SummEval across different evaluation aspects. The plots in Figure 5 show substantial variation in internal consistency across models and aspects, highlighting that some LLM judges exhibit significantly higher rates of transitivity violations than others. These results explain the performance gap between hard BT and soft BT, and motivate the need for reliability-aware aggregation when combining judgments from multiple LLMs.

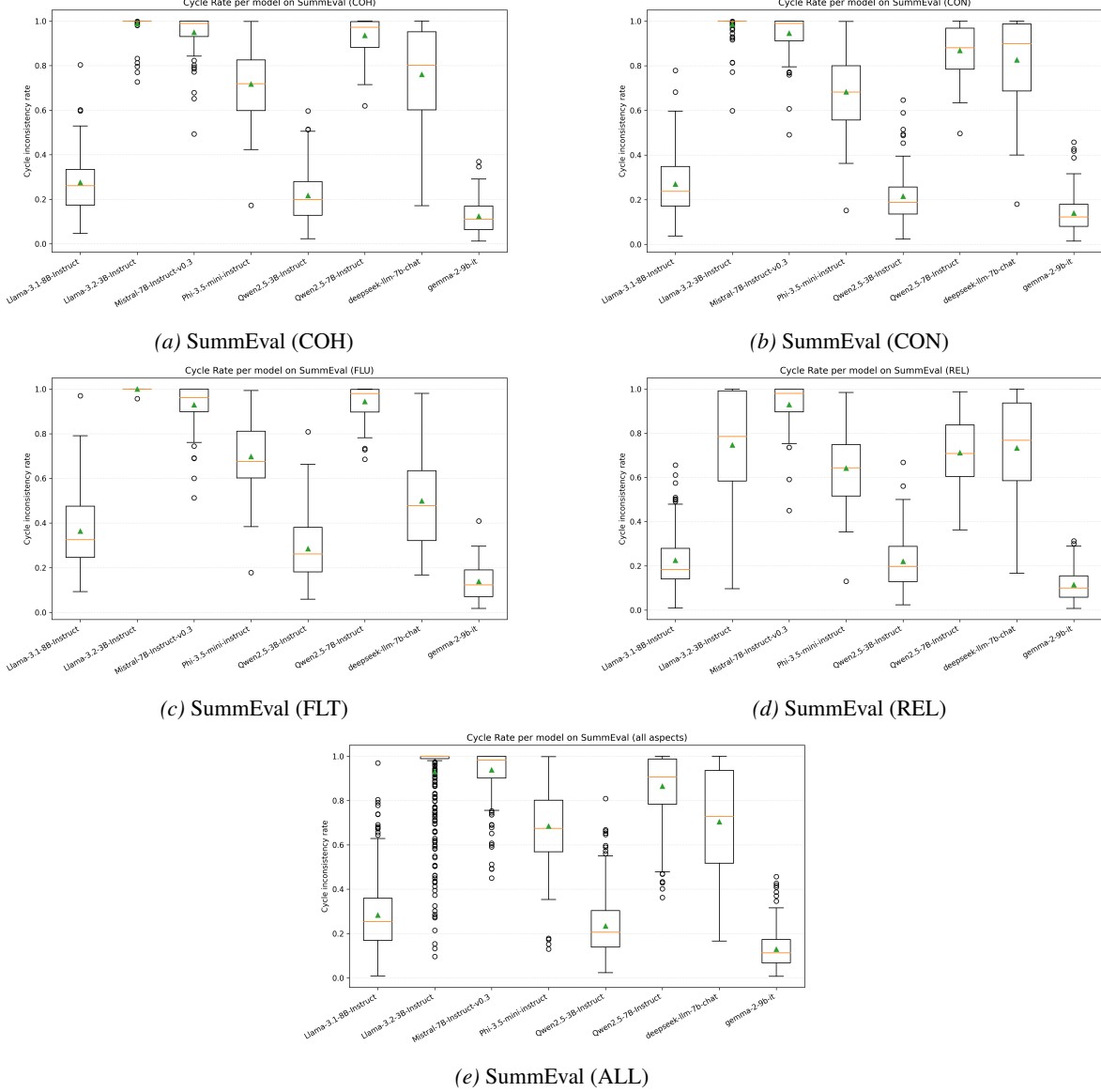

*(a)* SummEval (COH)  *(b)* SummEval (CON)

*(c)* SummEval (FLT)  *(d)* SummEval (REL)

*(e)* SummEval (ALL)

*Figure 5.* Cycle inconsistency rate on different LLMs, evaluated on SummEval.

## A.2. Scatter Plot for Judge Discriminators on SummEval

Figure 6 visualises the relationship between the learned discriminator $1/\sigma_k$ and cycle inconsistency on SummEval. Each point represents an LLM judge. The strong positive correlation (PCC=90.3%) indicates that judges assigned lower reliability tend to exhibit higher cycle inconsistency, supporting the interpretation of $1/\sigma_k$ as a measure of judge reliability.

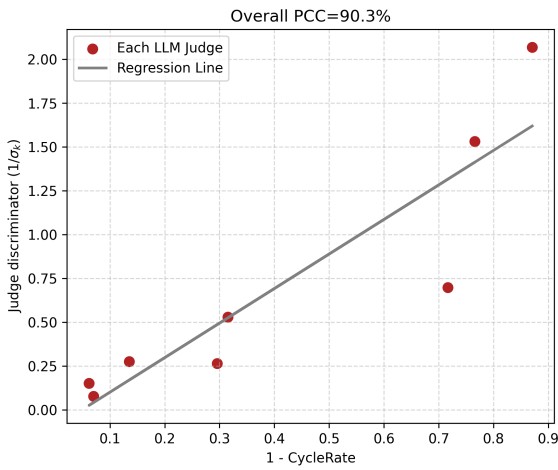

*Figure 6.* Scatter plot of $1/\sigma_k$ and $1 - \text{CycleRate}$ on SummEval.

## B. Additional Results on Topical-Chat

### B.1. Bradley-Terry Performance on Topical-Chat

The results in Table 5 show that the relative performance of hard BT and soft BT varies across judges and aspects, particularly for weaker or noisier models. Although soft BT surpasses hard BT in aggregation, it still underperforms on certain aspects, particularly ENG, suggesting a high level of inconsistency in LLM probabilities on this aspect.

*Table 5.* SRC of each individual LLM judge on Topical-Chat. The best results for each aspect on each model are underlined.

| LLM | Method | COH | CNT | ENG | NAT | ALL |
|---|---|---|---|---|---|---|
| Llama-3.1-8B | Avg-Prob | 51.82 | 44.06 | 62.61 | 46.90 | 51.34 |
| | hard BT | 49.65 | 41.87 | 61.92 | 51.70 | 51.29 |
| | soft BT | 50.57 | 45.92 | 58.73 | 49.62 | 51.21 |
| Llama-3.2-3B | Avg-Prob | 31.83 | 10.92 | 26.28 | 14.82 | 20.96 |
| | hard BT | 26.09 | 6.61 | 30.70 | 15.93 | 19.83 |
| | soft BT | 22.90 | 3.48 | 26.55 | 15.14 | 17.02 |
| Mistral-7B-Instruct-v0.3 | Avg-Prob | 33.98 | 33.17 | 46.67 | 27.67 | 35.37 |
| | hard BT | 49.51 | 43.71 | 58.11 | 42.08 | 48.35 |
| | soft BT | 40.36 | 34.37 | 54.45 | 32.37 | 40.39 |
| Phi-3.5-mini-instruct | Avg-Prob | 29.02 | 31.27 | 35.09 | 32.15 | 31.88 |
| | hard BT | 41.56 | 37.91 | 50.89 | 42.52 | 43.22 |
| | soft BT | 32.25 | 31.22 | 43.24 | 36.37 | 35.77 |
| Qwen2.5-3B-Instruct | Avg-Prob | 30.91 | 26.35 | 39.07 | 22.57 | 29.72 |
| | hard BT | 36.68 | 33.75 | 37.21 | 33.47 | 35.28 |
| | soft BT | 37.82 | 31.38 | 40.27 | 31.25 | 35.18 |
| Qwen2.5-7B-Instruct | Avg-Prob | 47.92 | 48.71 | 55.90 | 49.42 | 50.49 |
| | hard BT | 48.97 | 48.22 | 53.46 | 52.03 | 50.67 |
| | soft BT | 51.92 | 50.91 | 54.71 | 55.50 | 53.26 |
| DeepSeek-LLM-7B-Chat | Avg-Prob | 26.77 | 15.48 | 16.45 | 17.19 | 18.97 |
| | hard BT | 36.81 | 26.79 | 32.27 | 26.81 | 30.67 |
| | soft BT | 35.20 | 27.94 | 35.85 | 27.91 | 31.73 |
| Gemma-2-9B-IT | Avg-Prob | 53.95 | 52.92 | 67.24 | 52.13 | 56.56 |
| | hard BT | 64.37 | 57.88 | 71.14 | 61.23 | 63.66 |
| | soft BT | 57.86 | 55.34 | 69.21 | 55.29 | 59.42 |
| All models | Avg-Prob | 56.01 | 49.39 | 61.62 | 51.53 | 54.64 |
| | hard BT | 59.31 | 50.25 | 62.57 | 56.90 | 57.26 |
| | soft BT | 60.05 | 53.87 | 61.87 | 58.20 | 58.50 |

## B.2. Cycle Inconsistency Rate on Topical-Chat

Figure 7 shows the cycle inconsistency rates of individual LLM judges on Topical-Chat across different aspects. Inconsistency levels are generally higher than on SummEval, consistent with the observation in Section 5.1 that probability noise is more substantial on Topical-Chat.

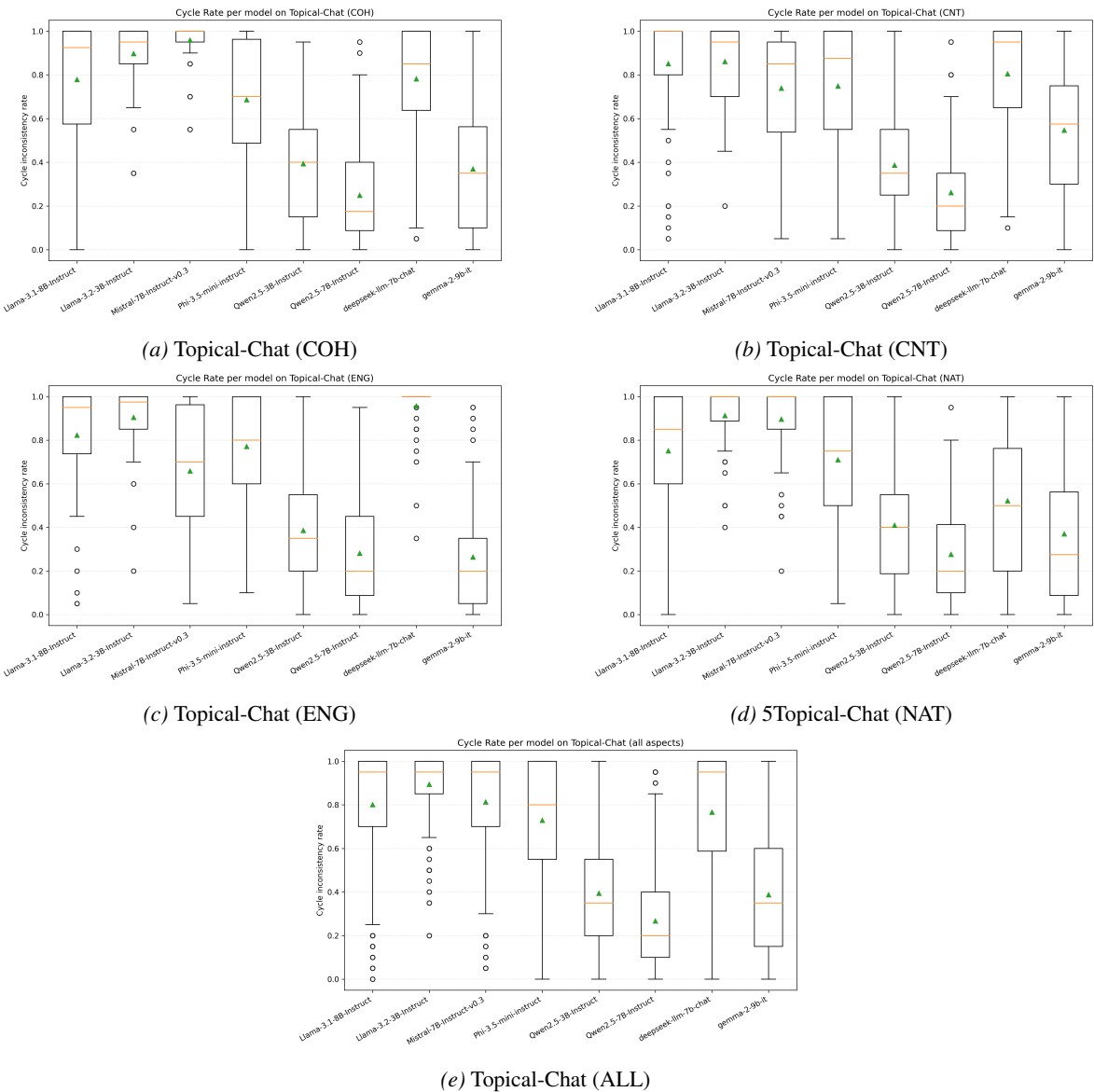

*(a)* Topical-Chat (COH)

*(b)* Topical-Chat (CNT)

*(c)* Topical-Chat (ENG)

*(d)* 5Topical-Chat (NAT)

*(e)* Topical-Chat (ALL)

*Figure 7.* Cycle inconsistency rate on different LLMs, evaluated on Topical-Chat.

## B.3. Scatter Plot for Judge Discriminators on Topical-Chat

Figure 8a plots $1/\sigma_k$ against the evaluation performance of each LLM judge on Topical-Chat, measured by SRC between Avg-Prob rankings and human judgements. A positive relationship is observed (overall PCC=67.4%), indicating that judges assigned smaller discriminator values (i.e. upweighted) tend to perform better individually, consistent with the interpretation of $1/\sigma_k$ as an unsupervised reliability signal. The correlation is somewhat lower than on SummEval (PCC=72.2%), which may reflect the higher overall inconsistency levels on Topical-Chat making individual judge performance harder to predict from discriminator values alone. Figure 8b plots $1/\sigma_k$ against $1 - \text{CycleRate}$ on Topical-Chat. A strong positive correlation is observed (overall PCC=93.1%), consistent with the SummEval results (PCC=90.3%), confirming that judges assigned

higher reliability by BT-$\sigma$ tend to exhibit fewer transitivity violations. This strong and consistent correlation across both datasets supports the interpretation of $\sigma_k$ as a principled unsupervised measure of judge internal consistency.

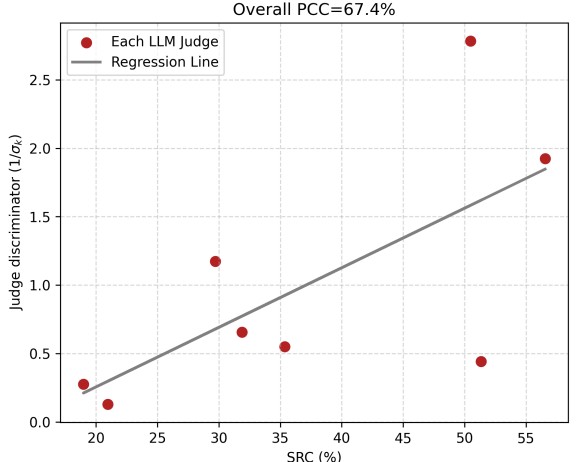
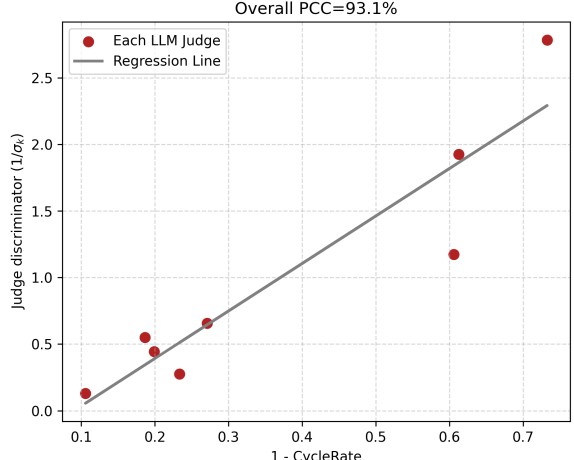

*(a)* Scatter plot of $1/\sigma_k$ and LLM performance with Avg-Prob, measured by SRC on Topical-Chat (ALL).

*(b)* Scatter plot of $1/\sigma_k$ and $1-$CycleRate, on Topical-Chat (ALL).

*Figure 8.* Scatter plots of judge discriminators on Topical-Chat.

### B.4. Jury Size Ablation on Topical-Chat

Table 6 reports aggregation performance (overall SRC) on Topical-Chat for jury sizes ranging from 2 to 8 judges. For each jury size, a representative subset of judges is selected from the full panel of 8. Across all jury sizes, BT-$\sigma$ and hard BT-$\sigma$ consistently outperform Avg-Prob and standard BT variants. BT-$\sigma$ remains effective even with as few as 2 judges (52.99 overall SRC), outperforming soft BT (50.91) and Avg-Prob (46.41). This suggests that meaningful reliability-aware aggregation is possible even with a minimal jury. Performance of BT-$\sigma$ is also robust to judge removal. Reducing from 8 to 2 judges results in a 6.93 point drop for BT-$\sigma$ (59.92 to 52.99) compared to an 8.23 point drop for Avg-Prob (54.64 to 46.41). These results suggest that BT-$\sigma$ benefits from additional judges but does not rely on a particular judge or jury size to be effective.

*Table 6.* Overall SRC on Topical-Chat for different jury sizes.

| # Judges | Avg-Prob | hard BT | soft BT | BT-$\sigma$ | hard BT-$\sigma$ |
|---|---|---|---|---|---|
| 8 | 54.64 | 57.26 | 58.50 | 59.92 | **60.15** |
| 7 | 49.81 | 52.99 | 53.99 | **55.13** | 52.76 |
| 6 | 53.24 | 56.63 | 58.49 | **59.46** | 56.82 |
| 5 | 53.46 | 57.20 | 56.66 | **59.51** | 55.77 |
| 4 | 55.27 | 58.85 | 58.53 | 59.09 | **63.27** |
| 3 | 55.64 | 58.56 | 58.96 | 58.90 | **63.47** |
| 2 | 46.41 | 48.83 | 50.91 | **52.99** | 50.51 |

## C. Generalisation on NovelEval

### C.1. Bradley-Terry Performance on NovelEval

Table 7 and 8 report SRCs of each individual LLM judge and aggregated methods on a subset of NovelEval (Sun et al., 2023), evaluated on the relevance (REL) aspect. We use a subset of 7 out of 21 context ids, excluding queries where Avg-Prob SRC

is near zero across all judges, which indicates near-random LLM judgements rather than a failure of the aggregation method. Results are consistent with the main findings on SummEval and Topical-Chat. For individual judges, hard BT and soft BT generally outperform Avg-Prob. In the aggregated setting, BT-$\sigma$ achieves the best overall SRC of 46.92, outperforming soft BT (45.28) and Avg-Prob (43.59) by 1.64 and 3.33 points respectively.

*Table 7.* SRC on NovelEval (subset of 7 queries) with individual LLM judges.

| LLM | Avg-Prob | hard BT | soft BT |
|---|---|---|---|
| Llama-3.1-8B | 40.30 | 42.11 | 44.94 |
| Llama-3.2-3B | 18.52 | 27.15 | 23.61 |
| Mistral-7B-Instruct-v0.3 | 39.01 | 42.62 | 41.42 |
| Phi-3.5-mini-instruct | 39.85 | 41.83 | 42.71 |
| Qwen2.5-3B-Instruct | 42.90 | 43.87 | 41.80 |
| Qwen2.5-7B-Instruct | 43.78 | 43.72 | 43.78 |
| deepseek-llm-7b-chat | 25.82 | 28.92 | 33.56 |
| gemma-2-9b-it | 44.64 | 45.16 | 45.33 |

*Table 8.* SRC on NovelEval (subset of 7 queries) with aggregated approaches.

| Method | REL |
|---|---|
| Avg-Prob | 43.59 |
| hard BT | 43.42 |
| soft BT | 45.28 |
| BT-$\sigma$ | **46.92** |
| hard BT-$\sigma$ | 45.69 |

## C.2. Plots on NovelEval

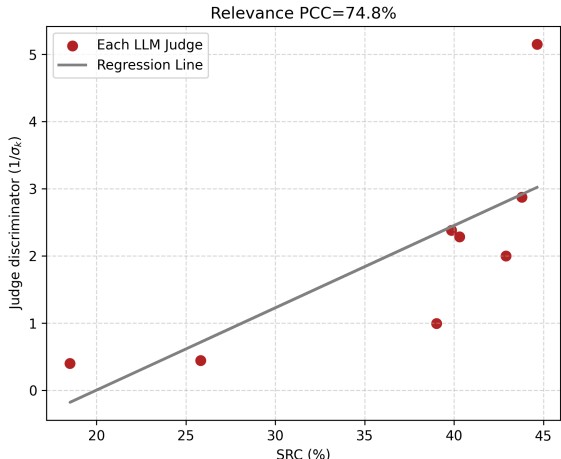

*Figure 9.* Scatter plot of $1/\sigma_k$ and LLM performance with Avg-Prob, measured by SRC on NovelEval (Relevance).

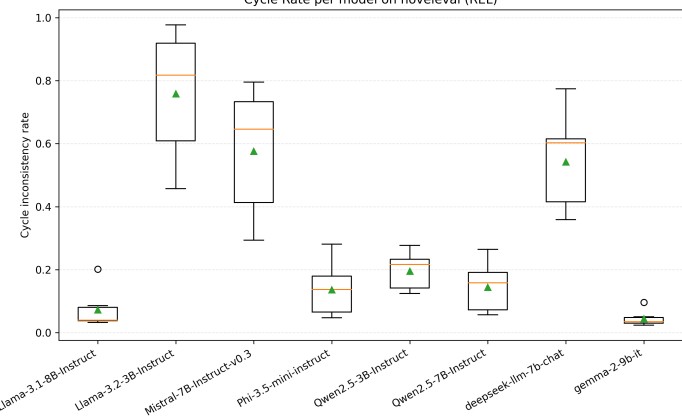

*Figure 10.* Cycle inconsistency rate on different LLMs, evaluated on NovelEval (Relevance)

## D. LLM Judge and Prompts

We use a diverse set of instruction-tuned large language models (LLMs) as judges to perform pairwise comparative assessment. The models include `Llama-3.1-8B-Instruct` (Grattafiori et al., 2024), `Llama-3.2-3B-Instruct` (Grattafiori et al., 2024), `Mistral-7B-Instruct-v0.3` (Jiang et al., 2023), `Phi-3.5-mini-instruct` (Microsoft, 2024), `Qwen2.5-3B-Instruct` (Yang et al., 2025), `Qwen2.5-7B-Instruct` (Yang et al., 2025), `DeepSeek-LLM-7B-Chat` (DeepSeek-AI, 2024), and `Gemma-2-9B-IT` (Team et al., 2024). These models vary in size, architecture, and training data, providing a heterogeneous pool of judges. For each dataset and evaluation aspect, LLMs are prompted to compare two candidate items (i.e. summaries for the same article or responses for the same dialogue context) and answer which candidate is better in the given metric. Preference probabilities are derived from the model logits corresponding to the tokens *A* and *B*. No human labels are used to calibrate or tune the LLM judges; human annotations are used solely for evaluation.

The prompts used for the two datasets are shown below:

---

**SummEval Prompt**

Article: {text}
Summary A: {passage_i}
Summary B: {passage_j}
Which Summary is more {metric}? Output the letter A or B directly.
Your output:

---

**Topical-Chat Prompt**

Context: {text}
Response A: {passage_i}
Response B: {passage_j}
Definition of {metric}: {metric_description}
Which response is more {metric}? Output the letter A or B directly.
Your output:

---

