# OpenReview forum: "Who can we trust? LLM-as-a-jury for Comparative Assessment"
_ICML.cc/2026/Conference — ICML 2026 regular_

### Official Review · Reviewer_2iMg · 2026-02-15

**Soundness:** 3
**Presentation:** 3
**Significance:** 3
**Originality:** 3
**Overall Recommendation:** 4
**Confidence:** 3

**Summary:**

This paper addresses the problem of aggregating pairwise comparative judgments from multiple LLM judges whose reliability varies. The authors first empirically demonstrate that LLM-generated pairwise probabilities often violate transitivity (cycle inconsistency), which explains why soft Bradley-Terry (BT) models can underperform hard BT on individual judges. They then propose BT-σ, an extension of the soft BT model that introduces a judge-specific discriminator parameter σ_k, jointly learned with item skill parameters from pairwise comparisons alone — without human labels. Experiments on SummEval and Topical-Chat show BT-σ consistently outperforms averaging-based aggregation and even supervised temperature scaling. The learned σ_k values correlate strongly with cycle inconsistency rates, providing an interpretable unsupervised signal of judge reliability.

**Compliance With Llm Reviewing Policy:**

Affirmed.

**Final Justification:**

The author's reply basically solved my problem, and I'm considering maintaining my positive rating.

**Key Questions For Authors:**

1、How does BT-σ perform with stronger or mixed-capability judges? The current jury is homogeneous in scale. Would adding one strong judge (e.g., GPT-4) change the dynamics significantly?

2、How sensitive is BT-σ to jury size and composition? What is the minimum number of judges needed for BT-σ to provide meaningful improvements? How does performance degrade if 1-2 judges are removed?

3、Can σ_k be item-dependent? The current formulation assumes each judge has a global reliability level. But a judge might be reliable for easy comparisons and unreliable for hard ones. Have you considered item-pair-specific or difficulty-aware extensions?

4、How do you ensure comparability of probabilities across architecturally different models? The logit-based extraction of yes/no probabilities may not yield commensurable signals across model families. Have you validated this assumption?

**Limitations:**

The paper is honest about its scope but several limitations deserve explicit acknowledgment. The evaluation is confined to two small NLG datasets with exhaustive pairwise comparisons — a setting that is computationally expensive and may not reflect practical deployment where comparisons are sparse. The jury consists only of small open-weight models, leaving open questions about applicability to mixed-scale or proprietary judge panels.

**Strengths And Weaknesses:**

Strengths

S1: Clean problem formulation and principled approach. The paper identifies a clear, well-scoped problem — heterogeneous judge reliability in LLM-as-a-jury settings — and proposes an elegant, minimal extension to the Bradley-Terry model. Adding one parameter per judge is simple, interpretable, and well-motivated by the connection to temperature scaling.

S2: Strong diagnostic analysis. The cycle inconsistency analysis (Section 5.1, Figure 2) is a valuable empirical contribution on its own. The observation that hard BT acts as a robust estimator under high inconsistency while soft BT benefits from moderate inconsistency is insightful and well-supported by data.

S3: Interpretable learned parameters. The strong correlation between 1/σ_k and both judge performance (PCC=72.2%) and cycle consistency (PCC=90.3%) on SummEval is compelling. This provides genuine interpretability — the model isn't just a black box that improves numbers; it learns something meaningful about each judge.

Weaknesses

W1: All judges are small open-weight models (3B–9B). The jury consists entirely of models in the 3B–9B range. It is unclear whether the same inconsistency patterns and BT-σ improvements hold when using stronger judges (e.g., GPT-4, Claude, Llama-70B) or when mixing small and large judges. In practice, many jury setups combine models of very different capability levels, and this is where reliability-aware aggregation should matter most.

W2: Marginal improvements in absolute terms. While BT-σ consistently outperforms baselines, the improvements are often small. On SummEval, BT-σ achieves 50.50% overall SRC vs. 49.40% for soft BT — a 1.1 percentage point gain. On Topical-Chat, the gain is 59.84% vs. 58.50% (1.34 points). These differences, while consistent, are modest. No statistical significance tests (e.g., bootstrap confidence intervals, paired tests across articles) are provided to confirm these differences are reliable.

W3: Unclear origin of LLM judge capability and probability estimates. The paper derives pairwise preference probabilities from "the model logits corresponding to the tokens yes and no" (Appendix C), but provides no analysis of whether this extraction method is well-calibrated or consistent across architecturally different models. Different LLMs may tokenize "yes"/"no" differently, and the logit gap between these tokens may not carry comparable probabilistic meaning across model families (e.g., Llama vs. Qwen vs. Gemini). The paper treats all judges' probability outputs as commensurable inputs to the BT model without validating this assumption.

---

> ### Author Rebuttal · Authors · 2026-03-31
>
> We thank Reviewer 2iMg for the positive comments and constructive feedback. We address each concern and question in detail below.
>
> **Weakness1/Question1: stronger or mixed-capability judges.**
>
> We thank the reviewer for this important question. Our current jury is not strictly homogeneous in capability. As shown in Table 1, judge performance **varies substantially** (e.g.,  large gaps in SRC across models from 17.84 to 46.83 on SummEval).  BT-σ already operates over this range of performance and assigns smaller discriminators $σ_k$ (higher reliability) to more consistent judges, increasing their influence in the aggregated ranking. The BT-σ framework itself is model-agnostic and naturally extends to mixed-capability settings. A stronger judge that produces more internally consistent comparisons would be assigned a smaller $σ_k$ and thus receive greater influence in the aggregation. Compared to simple averaging, which treats all judges equally, BT-σ provides a principled way to account for differences in reliability. We will clarify this behaviour in the paper.
>
>
> **Weakness2: marginal improvements and lack of significance testing.**
>
> We thank the reviewer for this suggestion. We have conducted statistical significance tests (p < 0.05 and p < 0.001) comparing each method against Avg-Prob. On SummEval, BT-σ shows statistically significant improvements on all aspects (p < 0.001 on COH and CON, and p < 0.05 on FLU and REL). BT-σ-asp and hard BT-σ variants also show consistent significance.  Results are more mixed on Topical-Chat: significance is reached on NAT (p < 0.05), while other aspects show consistent but not always significant gains. This aligns with our observation that Topical-Chat exhibits higher inconsistency, making aggregation more difficult.
> Beyond performance gains, BT-σ offers two additional contributions. First, the learned discriminator $σ_k$ provides an unsupervised and interpretable signal of judge reliability, strongly correlated with independent performance measures (Table 3, PCC > 72%). Second, our analysis provides a mechanistic explanation of when soft vs. hard BT succeeds by linking their performance gap directly to probability inconsistency.  These insights provide a principled understanding of aggregation behaviour beyond incremental improvements.
>
>
> **Weakness3/Question4: comparability of probabilities across model families.**
>
> We thank the reviewer for raising this question. We partially address this via symmetrisation (Eq. 3), which removes positional bias within each judge. More fundamentally, BT-σ does not require probabilities to be directly comparable across judges. The model relies on how well each judge's comparisons align with a shared global ranking structure, rather than on the absolute scale of their probabilities. The discriminator $σ_k$ captures this alignment by acting as a per-judge scaling parameter. Judges whose probabilities are less consistent with a coherent ranking, whether due to miscalibration or structural inconsistency, receive larger $σ_k$ values and are automatically downweighted during aggregation.
>
>
> **Question2: sensitivity to jury size and composition.**
>
> We evaluate jury sizes from 2 to 8 judges on Topical-Chat.
> Across all jury sizes, BT-σ and hard BT-σ consistently outperform Avg-Prob and standard BT, including with as few as 2 judges (BT-σ: 52.99 vs. soft BT: 50.91 vs. Avg-Prob: 46.41 overall SRC). This suggests that meaningful reliability-aware aggregation is possible even with a minimal jury.
> Performance of BT-σ is also robust to judge removal. Reducing from 8 to 2 judges results in only a modest change in BT-σ overall SRC (59.92 to 52.99). This suggests that BT-σ benefits from additional judges but does not rely on a particular judge or a particular number of judges to be effective. We will include this ablation in the paper.
>
>
> **Question3: item-pair-specific or difficulty-aware $σ_k$**
>
> Extending $σ_k$ to be pair-specific is straightforward to implement, but its interpretation within the BT framework becomes unclear. In BT, item skills are defined in a global latent space, and $σ_k$ captures how consistently a judge's comparisons align with this shared ranking structure. Making reliability pair-specific assigns a separate parameter to each comparison, which no longer reflects global consistency but instead absorbs local noise.
> A more natural extension is to model reliability at a coarser granularity (e.g. per-sample or per-aspect), where consistency can still be defined relative to a shared structure. Our BT-σ-asp variant, which estimates judge-aspect-specific discriminators, explores this and yields only modest improvements, suggesting that judge reliability is relatively stable across aspects in practice.
> We are also exploring methods to detect and filter unreliable comparisons prior to aggregation, these extensions are considered complementary to BT-σ and beyond the scope of the current paper. We will clarify this discussion in the revision.

---

> > ### Author Rebuttal · Reviewer_2iMg · 2026-04-02
> >
> > Thank you for your reply. I look forward to your final revised version.

---

### Official Review · Reviewer_Ed4k · 2026-02-20

**Soundness:** 3
**Presentation:** 3
**Significance:** 3
**Originality:** 3
**Overall Recommendation:** 4
**Confidence:** 2

**Summary:**

This work addresses a critical limitation of the LLM-as-a-judge paradigm in reference-free NLG evaluation: the inherent heterogeneity and probabilistic inconsistency across LLM judges, a gap where existing methods either assume uniform reliability or rely on impractical supervised calibration with human annotations. To this end, the authors propose BT-$\sigma$, an unsupervised reliability-aware aggregation model integrated within an LLM-as-a-jury framework. Extending the Bradley-Terry model, BT-$\sigma$ introduces a discriminator-specific parameter $\sigma_k$ for each judge to explicitly quantify its reliability and discriminative power, enabling the joint inference of candidate rankings and judge weights directly from pairwise comparison data without human labels. Complementary variants are presented, including hard BT-$\sigma$ for scenarios with severe probabilistic inconsistency, BT-$\sigma$-asp for aspect-specific calibration, and a symmetrization step to mitigate positional bias, with comprehensive validation conducted on both the SummEval and Topical-Chat datasets. Substantively, this study makes three key contributions: it provides the first systematic empirical diagnosis and quantification of inconsistency in LLM pairwise judgments; it pioneers the development of an unsupervised aggregation scheme for modeling multi-judge reliability in this setting; and it validates that the learned $\sigma_k$ parameters serve as a robust unsupervised proxy for judge quality. Collectively, these advances establish a principled and practical solution to enhance the robustness and interpretability of automatic NLG evaluation pipelines.

**Compliance With Llm Reviewing Policy:**

Affirmed.

**Final Justification:**

My final recommendation is weak accept. The authors’ rebuttal has addressed my concerns, and I will keep my original score.

**Key Questions For Authors:**

1. This paper only quantifies the impact of third-order cyclic inconsistency on LLM judgments. Have the authors attempted to conduct a quantitative analysis of higher-order ($k\geq4$) cyclic inconsistency? Does higher-order cyclic inconsistency have a significant impact on the aggregation performance of the BT-$\sigma$ model? If higher-order inconsistency is considered, is any corresponding adjustment required for the design of the BT-$\sigma$ model?

2. The paper only makes a basic comparison of the mathematical relationship between $\sigma_k$ and supervised temperature scaling without in-depth analysis. Could the authors provide the rigorous mathematical derivational relationship and transformation boundary conditions between the two? What are the fundamental technical differences between the unsupervised calibration based on $\sigma_k$ and traditional supervised temperature scaling?

3. The paper only validates the proposed method on two public datasets. Is the method in this paper applicable to a broader range of datasets?

4. Reliability estimation may introduce an additional risk: if an LLM jury exhibits systematic biases on a certain issue, with only a small number of judges capable of making accurate assessments, these few correctly performing judges may be assigned lower reliability. This method appears to encourage a form of herd mentality, in which the minority of judges are required to conform to the majority. How do the authors propose to address this issue?
\end{enumerate}

**Limitations:**

yes

**Strengths And Weaknesses:**

This paper addresses the critical challenge of heterogeneity and inconsistency across multiple judges in the LLM-as-a-judge paradigm for reference-free NLG evaluation, proposing the BT-$\sigma$ model for unsupervised reliability-aware aggregation.

Soundness: The theoretical derivations are logically consistent, and the design of BT-$\sigma$, bias mitigation strategies, and inconsistency quantification methods are well-founded. The experiments adopt canonical datasets and appropriate metrics, with comprehensive baseline comparisons that effectively validate the core claims. The authors have also objectively disclosed the model's limitations. However, only third-order cyclic inconsistency is quantified, the analysis of the mathematical relationship between $\sigma_k$ and temperature scaling remains superficial, validation is limited to merely two datasets, and the model's performance in broader domains remains unknown.

Presentation: The manuscript is well-structured and logically coherent. The research context is clearly contextualized, with a precise delineation of differences from existing work. Experimental details are sufficiently comprehensive to enable reproducibility, and the figures/tables are highly complementary to the main text.

Significance: This work directly targets a core pain point of the LLM-as-a-judge paradigm, filling the gap in unsupervised reliability modeling for multi-judge aggregation in reference-free NLG evaluation. The proposed BT-$\sigma$ model eliminates the need for human annotations, offering strong practical utility. Its key findings provide important guidance for future research, and the core modeling philosophy is generalizable to other fields involving pairwise comparisons.

Originality: This study pioneers the systematic investigation of unsupervised reliability aggregation for multiple LLM judges, quantifying the impact of LLM judgment inconsistency. The introduction of the $\sigma_k$ parameter and the use of cyclic inconsistency rate provide the community with novel methodological tools and evaluation metrics for this setting.

---

> ### Author Rebuttal · Authors · 2026-03-31
>
> We thank the reviewer for the constructive and detailed feedback. We have incorporated additional experiments and analysis to address the concerns and respond to each point below.
>
> **Question1: higher-order (k ≥ 4) cyclic inconsistency**
>
> In our work, cycle inconsistency rate is used as a diagnostic metric to quantify internal inconsistency in LLM judgments. We focus on 3-cycles (k=3) as they provide the simplest and most interpretable form of transitivity violation, and are commonly used as a proxy for inconsistency in prior work (Wang et al., 2025). Higher-order cycles (k≥4) can be **decomposed** into 3-cycles, making 3-cycle analysis a representative measure of overall inconsistency. Crucially, BT-σ does not depend on the order of inconsistency and requires no modification to handle higher-order cycles, as it operates directly on pairwise comparisons and captures reliability through deviations from a globally consistent BT structure. Empirically, the learned discriminator σ shows strong correlation with 3-cycle rates (Table 4, PCC > 90%). To further validate this, we compute 4-cycle rates on SummEval and observe similar patterns, i.e. a strong correlation between 4-cycle rates and judge reliability. We will clarify this point in the revision.
>
> **Question2: mathematical relationship between $σ_k$ and temperature scaling**
>
> The relationship between $σ_k$ and temperature scaling is reflected in Eqs. 5-7 of the paper, which we will clarify further.
>
> For a single judge, temperature scaling with parameter T transforms probabilities as $\tilde{p}\_{ij} = \sigma \left(\frac{s_i - s_j}{T} \right)$. The discriminator $σ_k$ plays the same role by rescaling the skill gap before applying the logistic function, and is therefore mathematically equivalent to a per-judge temperature.
> However, this equivalence holds only when the judge's pairwise probabilities are perfectly consistent with the BT model, i.e., $p_{ij} = \sigma(s_i - s_j)$ for some latent skills. In this case, both temperature scaling and $σ_k$ reduce to a simple rescaling of the skill space and do not affect the induced ranking.
>
> In practice, LLM probabilities are not perfectly consistent. Under such conditions, the two approaches diverge fundamentally. Temperature scaling is a supervised, post-hoc calibration method that adjusts probability magnitudes to match human labels but does not resolve structural inconsistencies such as transitivity violations. In contrast, BT-σ is an unsupervised model that jointly infers item skills and $σ_k$ by fitting a globally consistent BT structure to the observed comparisons, where $\sigma_k$ absorbs judge-specific noise and inconsistency rather than merely rescaling outputs.
> This distinction explains our empirical results in Table 2 that BT-σ consistently outperforms supervised Temp-BT, as it addresses structural inconsistency rather than only magnitude miscalibration.
>
> **Question3: datasets**
>
> We thank the reviewer for raising this point. SummEval and Topical-Chat are standard benchmarks for NLG ranking evaluation and remain widely used in prior work.
> To assess generalisation, we conducted additional experiments on a subset of NovelEval (Sun et al., 2023), following Liu et al. (2024). We had to use a subset of NovelEval since there are many samples where all the LLMs are making random judgements. We excluded samples where **Avg-Prob** SRC was near zero indicating random judgements. On the retained subset, results are consistent with our main findings (see Table below): BT-σ achieves the best overall SRC (46.92), outperforming soft BT (45.28) and Avg-Prob (43.59). The pattern across individual judges also aligns with our earlier observations: BT-based methods consistently outperform Avg-Prob, and reliability-aware aggregation yields additional gains. These results support the generalisation of our approach beyond the original benchmarks. We will include full results in the revision.
>
> |Method|Relevance|
> |-|-|
> |Avg-Prob|43.59|
> |hard BT|43.42|
> |soft BT|45.28|
> |BT-σ|__46.92__|
> |hard BT-σ|45.69|
>
> **Question4: herd mentality**
>
> We thank the reviewer for raising this important point. This limitation is shared by all unsupervised aggregation methods. Without ground truth, it is not possible to distinguish a systematically biased majority from a correct minority. Importantly, BT-σ differs from simple averaging in how reliability is estimated. Simple averaging implicitly treats agreement with the majority as a signal of reliability, whereas BT-σ estimates reliability based on each judge's internal consistency (e.g., transitivity). A highly consistent minority judge can therefore be upweighted even if it disagrees with the majority, while inconsistent judges are downweighted regardless of majority alignment.
> If the majority is both biased and internally consistent, this bias may still be reinforced, which is a fundamental limitation of any unsupervised aggregation. We will clarify this in the limitations section.

---

> > ### Author Rebuttal · Reviewer_Ed4k · 2026-04-01
> >
> > Thanks to the authors for their explanations.

---

### Official Review · Reviewer_N8vF · 2026-03-12

**Soundness:** 3
**Presentation:** 3
**Significance:** 3
**Originality:** 3
**Overall Recommendation:** 4
**Confidence:** 3

**Summary:**

This paper studies how to aggregate pairwise judgments from multiple LLM judges for comparative assessment when the judges have heterogeneous reliability. The authors propose BT-σ, a judge-aware extension of the Bradley-Terry model that jointly learns item rankings and judge-specific reliability parameters from pairwise comparisons alone.

**Compliance With Llm Reviewing Policy:**

Affirmed.

**Final Justification:**

My concerns about the originality are largely resolved through the authors' rebuttal. I adjusted my score accordingly.

**Key Questions For Authors:**

* The experiments appear to focus on evaluations along a single aspect of a sentence. In practice, however, LLM judges often evaluate outputs with multiple aspects simultaneously. Would the proposed method remain effective in such multi-aspect evaluation settings?
* Could the authors clarify the baseline methods used in the experiments, especially AvgProb?
* How much computational cost does the proposed multi-judge framework require? In particular, how does the cost scale with the number of judges and pairwise comparisons?

**Limitations:**

yes

**Strengths And Weaknesses:**

[Strengths]
* The reliability estimation of each LLM-as-a-Judge without human labels is an important and challenging problem for reliable evaluation.
* The proposed method is well motivated, and its formulation and derivation are reasonable.

[Weaknesses]
* The paper lacks sufficient implementation details for reproducibility, such as hyperparameter settings and other experimental details.
* The novelty relative to prior work is not entirely clear. Bradley-Terry-based methods for estimating annotator reliability have been widely studied in the crowdsourcing literature, and there are closely related methods such as Crowd-BT (Chen et al., “Pairwise Ranking Aggregation in a Crowdsourced Setting,” WSDM 2013). The paper should more clearly explain how the proposed method differs from and improves upon these prior approaches.

---

> ### Author Rebuttal · Authors · 2026-03-31
>
> We thank Reviewer N8vF for the constructive and insightful feedback. We have incorporated additional experiments, clarified the experimental setup, and strengthened the analysis to address the concerns. We respond to each point below.
>
> **Weakness1: reproducibility.**
>
> We apologise for the lack of explicit detail and will add a dedicated implementation subsection.
> Specifically, BT-σ is optimised by maximising the joint log-likelihood over item skills {$s$} and judge discriminators {$\sigma_k$}:
> $$
> \mathcal{L}(s, \{\sigma_k\}) \propto \sum_{k=1}^{K} \sum_{(i,j)\in C} \left[ p^{(k)}_{ij} \log \sigma\left(\frac{s_i - s_j}{\sigma_k}\right) + (1 - p^{(k)}\_{ij}) \log \left(1 - \sigma\left(\frac{s_i - s_j}{\sigma_k}\right)\right) \right].
> $$
> We use the minimize function from SciPy with the L-BFGS-B method, which typically converges within ~100 iterations. Both $\sigma_k$ and $s_i$ are initialised with random values drawn from a uniform distribution over [0, 1).  This optimiser adapts step sizes internally via a quasi-Newton Hessian approximation instead of using a learning rate. We will also make our code available to support reproducibility.
>
>
> **Weakness2: Crowd-BT.**
>
> We thank the reviewer for raising this important point. Crowd-BT (Chen et al., 2013) is indeed a related method, and we will add an explicit comparison in the revision. However, BT-σ differs from Crowd-BT in a fundamental way. Crowd-BT models annotator reliability via a mixture parameter $η_k$, where the predicted preference is:
> $$
> \hat{P}(z_i \succ z_j | s_i, s_j, y^{(k)}\_{ij}, \eta\_k ) = \tilde{p}^{(k)}\_{ij} \sigma(s_i - s_j) + (1 - \tilde{p}^{(k)}\_{ij})(1 - \sigma(s_i - s_j)),
> $$
> where $\tilde{p}^{(k)}\_{ij} = \eta_k$ ​ if $y^{(k)}\_{ij} = 1$ and $1 - \eta_k$ otherwise. Here $ η_k$ represents the probability that worker $k$ makes the correct binary decision, a mixture between a reliable and a random annotator.
> BT-σ instead introduces $\sigma_k$ as a discriminator that directly rescales the latent skills:
> $$
> \hat{P}(z_i \succ z_j | s_i, s_j, \sigma_k) = \sigma\left(\frac{s_i - s_j}{\sigma_k}\right),
> $$
> $\sigma_k$ modulates how sharply a judge discriminates between items, this is not a mixture model. Critically, Crowd-BT is restricted to binary decisions $y^{(k)}\_{ij} \in \\{0,1\\}$, whereas BT-σ naturally handles both soft preference probabilities $p^{(k)}_{ij} \in [0,1]$ and binary decisions, and the latter corresponding to hard BT-σ in our framework.
>
> To validate this distinction empirically, we implemented Crowd-BT as an additional baseline. Results on SummEval (see Table below) show that BT-σ (overall SRC=50.50) outperforms Crowd-BT (SRC=48.35) and soft BT (SRC=49.40), demonstrating that BT-σ is better suited to the LLM jury setting than the Crowd-BT approach. This observation is consistent across each evaluation aspect. We will include Crowd-BT in the related work and incorporate it as a baseline in the revision.
>
> | Method | ALL |
> |--|--|
> | Avg-Prob | 45.15 |
> | hard BT | 47.34 |
> | soft BT | 49.40 |
> | Crowd-BT | 48.35 |
> | Temp-BT | 50.16 |
> | BT-σ | 50.50 |
> | BT-σ-asp | **50.65** |
> | hard BT-σ | 48.31 |
>
>
> **Question1: multi-aspect evaluation.**
>
> The reviewer raises an important point. Our current setup follows standard NLG evaluation practice, where LLM judges are prompted per aspect, yielding aspect-specific comparisons. This choice is also aligned with the available benchmarks, which provide annotations at the aspect level rather than a single holistic score. However, BT-σ does not rely on this assumption and can be directly applied to holistic judgments where multiple aspects are considered simultaneously and a single preference probability is produced. In this setting, the observed probabilities can be viewed as implicitly aggregating multiple aspects, and the learned discriminator reflects each judge's overall reliability. The trade-off is that aspect-specific interpretability is no longer available, as signals from different aspects are entangled.
>
>
> **Question2: clarification of AvgProb baseline.**
>
> Avg-Prob serves as the baseline in the work, it directly averages each candidate's win probability across all pairwise comparisons (i.e. $w_i = \frac{1}{N-1} \sum_{j \ne i}p_{ij}$, Equation in Section 5.1). This approach treats comparisons independently and does not enforce transitivity or any global ranking structure, in contrast to BT-based methods, which infer a consistent latent ranking. We will clarify this definition and its role as a non-structured baseline in Section 5.1.
>
>
> **Question3: computational cost.**
>
> BT-σ adds minimal overhead over the soft BT approach. The number of additional parameters is equal to the number of judges K (at most 8 in our experiments). For a representative setting (100 articles, ~240 comparisons per article, and 8 judges; ~192K comparisons in total), optimisation completes in **5-6 hours on a CPU**. The cost scales linearly with the number of judges and comparisons.

---

> > ### Author Rebuttal · Reviewer_N8vF · 2026-04-03
> >
> > Thank you for the detailed rebuttal. My concerns are now largely resolved. I'll adjust my score accordingly.

---

### Official Review · Reviewer_McoA · 2026-03-12

**Soundness:** 3
**Presentation:** 3
**Significance:** 3
**Originality:** 3
**Overall Recommendation:** 4
**Confidence:** 3

**Summary:**

The submission presents BT-σ, an extension of the Bradley-Terry model designed for "LLM-as-a-jury" scenarios. The core contribution is the design of a discrimination parameter for each LLM judge, allowing the model to automatically reweight individual reliability. BT-σ performs joint optimization of judge reliability and item skills in a fully unsupervised manner, bypassing the need for human labels. Through experiments, the authors empirically validate the necessity of establishing a global ranking and demonstrate that BT-σ effectively mitigates judge-level noise in pairwise comparative assessment tasks.

**Compliance With Llm Reviewing Policy:**

Affirmed.

**Final Justification:**

After carefully reading the authors’ rebuttal and considering the additional clarifications and experimental results they provided, I have decided to increase my overall score from Weak Reject to a  Weak Accept.

**Key Questions For Authors:**

1. Could you run your method on more pairwise comparison benchmarks? Please refer to Weakness 1 for details. You can conduct a lightweight experiment on a subset of any newer benchmark. If not possible or feasible in time, it’s also acceptable.
2. I am curious whether you can derive an explicit formulation (e.g., an optimization objective) for your proposed BT-σ at the end of the method section (before Section 4). In contrast, there are many formulas in the preceding introductory parts, which may make it difficult for readers to locate the core idea of your method.

**Limitations:**

yes

**Strengths And Weaknesses:**

Strength：
1. The paper is well-motivated, addressing the critical need for modeling relative reliability among multiple LLM judges. The proposed unsupervised mehod for reliability modeling is both practical and insightful, offering a valuable reference for future research.
2. The empirical analysis is solid. A key highlight is the analysis showing that the learned discrimination parameter σk correlates strongly with the independently measured "Cycle Inconsistency Rate". This provides convincing evidence that the model successfully captures the internal logical consistency of the judges.


Weakness：
1. Limited Evaluation Benchmarks. The experiments rely on SummEval and Topical-Chat, both of which are somewhat dated (from 2020 and 2021). The improvements on these datasets are minor and limit the persuasiveness of the proposed method. It remains unclear whether the method can generalize to more contemporary pairwise evaluation tasks like MT-Bench, JudgeBench, or RewardBench.
2. Reliance on Token Logits. This method requires access to model logits to derive preference probabilities. This restricts the "jury" to open-source models only, which is an inevitable limitation. Since closed-source Judges are assumed to be more accurate,  they are more likely to be adopted in evaluation tasks.
3. Potential for Systematic Bias. The unsupervised optimization may be susceptible to systematic biases. For instance, if the majority of judges in the jury share a common latent bias (e.g., favoring longer responses), the model might incorrectly interpret this consensus as high reliability. Or, if the jury consists of a majority of small-scale judges and a minority of large-scale (more capable) judges, will the biased opinions of the majority mislead the overall evaluation and assign smaller weights to accurate judges?

---

> ### Author Rebuttal · Authors · 2026-03-31
>
> We thank Reviewer McoA for the constructive and detailed feedback. We have incorporated additional experiments and analysis to address the concerns and respond to each point below.
>
> **Weakness1/Question1: Limited evaluation benchmarks.**
>
> We thank the reviewer for this suggestion. SummEval and Topical-Chat are standard benchmarks for NLG ranking evaluation and remain widely used in prior work. In contrast, MT-Bench, JudgeBench, and RewardBench primarily focus on pairwise or scalar assessment rather than recovering a global ranking, making direct comparison less straightforward. As our method is designed for ranking aggregation, we focus on benchmarks that support this setting.
>
> Following the suggestions, we conducted additional experiments on a subset of NovelEval (Sun et al., 2023), following prior work in this area (Liu et al., 2024). We had to use a subset of NovelEval since there are many samples where all the LLMs are making random judgements. We excluded samples where **Avg-Prob** SRC was near zero, indicating random judgements. Results are shown below:
>
> | Method           | Relevance |
> |------------------|-----------|
> |Avg-Prob         | 43.59     |
> |hard BT          | 43.42     |
> |soft BT          | 45.28     |
> |BT-σ         | __46.92__     |
> |hard BT-σ    | 45.69     |
>
> On NovelEval, results are consistent with our main findings: BT-σ achieves the best overall SRC of 46.92, outperforming soft BT (45.28) and Avg-Prob (43.59) by 1.64 and 3.33 points, respectively (see Table below). The pattern across individual judges also aligns with our earlier observation on SummEval and Topical-Chat: soft BT and hard BT consistently outperform Avg-Prob, and reliability-aware BT variant yields additional gains in aggregation. These results support the generalisation of our approach beyond the two original benchmarks.
>
> We will include these results together with individual model results and discussions in the revision. We would also be happy to evaluate our method on additional benchmarks for NLG ranking if the reviewer can suggest suitable datasets for this setting.
>
> **Weakness2: Reliance on Token Logits.**
>
> We thank the reviewer for raising this point. In our experiments, preference probabilities are derived from token-level logits for open-weight models, which provide a natural and well-defined probabilistic signal. However, the BT-σ framework itself is agnostic to how preference probabilities are obtained. It can be applied to any setting where pairwise preference estimates are available, including settings where probabilities are approximated via drawing multiple samples.
>
> We also note that using open-weight models enables **full reproducibility and controlled analysis** of judge reliability, which is difficult to guarantee with proprietary APIs where model behaviour may change over time. We will clarify this discussion in the paper.
>
>
> **Weakness3: Potential for Systematic Bias.**
>
> We thank the reviewer for raising this important point. This limitation is shared by __all unsupervised aggregation__ methods. Without ground truth, it is not possible to distinguish a systematically biased majority from a correct minority. Importantly, BT-σ differs from simple averaging in how reliability is determined. Simple averaging treats agreement with the majority implicitly as a signal of reliability, whereas BT-σ estimates reliability based on each judge's internal consistency (e.g., transitivity of its pairwise comparisons). A highly consistent minority judge can therefore be upweighted even if it disagrees with the majority, while internally inconsistent judges are downweighted regardless of majority alignment.
> If a majority of judges are both biased and internally consistent, this bias may still be reinforced, which is a fundamental limitation of unsupervised aggregation methods. We will make this explicit in the limitations section.
>
>
> **Question2: optimization objective.**
>
> We thank the reviewer for this helpful suggestion. We agree that presenting a clear optimisation objective would improve clarity and readability. In the revision, we will explicitly state the BT-σ objective at the end of the method section as the joint log-likelihood over all judges and comparisons:
> $$\mathcal{L}(s, \{\sigma_k\}) \propto \sum_{k=1}^{K} \sum_{(i,j)\in C} \left[ p^{(k)}_{ij} \log \sigma\left(\frac{s_i - s_j}{\sigma_k}\right) + (1 - p^{(k)}\_{ij}) \log \left(1 - \sigma\left(\frac{s_i - s_j}{\sigma_k}\right)\right) \right].$$
> We will also refine the presentation in the earlier parts of the Method Section to improve clarity and make the core idea easier to follow.

---

> > ### Author Rebuttal · Reviewer_McoA · 2026-04-06
> >
> > Thank you for the detailed responses and additional experiments. I will revise my score accordingly.

---

### Decision · Program_Chairs · 2026-04-30

**Decision:**

Accept (regular)

**Comment:**

The submission presents BT-σ, an extension of the Bradley-Terry model designed for "LLM-as-a-jury" scenarios. The core contribution is the design of a discrimination parameter for each LLM judge, allowing the model to automatically reweight individual reliability. BT-σ performs joint optimization of judge reliability and item skills in a fully unsupervised manner, bypassing the need for human labels. Through experiments, the authors empirically validate the necessity of establishing a global ranking and demonstrate that BT-σ effectively mitigates judge-level noise in pairwise comparative assessment tasks.

The paper is well-motivated and contains a solid, convincing empirical analysis. During the rebuttal period, the authors presented results on additional benchmarks; these should be included in the final version of the paper. Furthermore, the authors should add Crowd-BT to the related work section and include the promised comparison to their approach in the next revision.

Potential hallucinated reference (which should be addressed as well):
Reference: Wang, J., Liang, Y., Meng, F., Sun, Z., Shi, H., Li, Z., Xu, J., Qu, J., and Zhou, J. Is ChatGPT a Good Evaluator for Natural Language Generation? A Preliminary Study. arXiv:2303.07303, 2023.
Issue: Title mismatch with arXiv